# Cellular reprogramming with ATOH1, GFI1, and POU4F3 implicate epigenetic changes and cell-cell signaling as obstacles to hair cell regeneration in mature mammals

Amrita A Iyer[1], Ishwar Hosamani[1], John D Nguyen[2], Tiantian Cai[3], Sunita Singh[4], Melissa M McGovern[4], Lisa Beyer[5], Hongyuan Zhang[4], Hsin-I Jen[3,4†], Rizwan Yousaf[4‡], Onur Birol[3§], Jenny J Sun[4#], Russell S Ray[4], Yehoash Raphael[5], Neil Segil[2,6¶], Andrew K Groves[1,3,4*]

[1]Department of Molecular & Human Genetics, Baylor College of Medicine, Houston, United States; [2]Department of Stem Cell Biology and Regenerative Medicine, Keck School of Medicine of the University of Southern California, Eli and Edythe Broad Center for Regenerative Medicine and Stem Cell Biology at USC, Los Angeles, United States; [3]Program in Developmental Biology, Baylor College of Medicine, Houston, United States; [4]Department of Neuroscience, Baylor College of Medicine, Houston, United States; [5]Department of Otolaryngology-Head and Neck Surgery, University of Michigan, Ann Arbor, United States; [6]Caruso Department of Otolaryngology-Head and Neck Surgery, Keck School of Medicine of the University of Southern California, Los Angeles, United States

*For correspondence: akgroves@bcm.edu

Present address: †Ultragenyx, Cambridge, United States; ‡NIDCD, Bethesda, United States; §Georgia Institute of Technology, Atlanta, United States; #Moderna, Cambridge, United States

¶Deceased

**Abstract** Reprogramming of the cochlea with hair-cell-specific transcription factors such as ATOH1 has been proposed as a potential therapeutic strategy for hearing loss. ATOH1 expression in the developing cochlea can efficiently induce hair cell regeneration but the efficiency of hair cell reprogramming declines rapidly as the cochlea matures. We developed Cre-inducible mice to compare hair cell reprogramming with ATOH1 alone or in combination with two other hair cell transcription factors, GFI1 and POU4F3. In newborn mice, all transcription factor combinations tested produced large numbers of cells with the morphology of hair cells and rudimentary mechanotransduction properties. However, 1 week later, only a combination of ATOH1, GFI1 and POU4F3 could reprogram non-sensory cells of the cochlea to a hair cell fate, and these new cells were less mature than cells generated by reprogramming 1 week earlier. We used scRNA-seq and combined scRNA-seq and ATAC-seq to suggest at least two impediments to hair cell reprogramming in older animals. First, hair cell gene loci become less epigenetically accessible in non-sensory cells of the cochlea with increasing age. Second, signaling from hair cells to supporting cells, including Notch signaling, can prevent reprogramming of many supporting cells to hair cells, even with three hair cell transcription factors. Our results shed light on the molecular barriers that must be overcome to promote hair cell regeneration in the adult cochlea.

## Editor's evaluation

This study uses well-designed genetic approaches to examine how specific transcription factors regulate hair cell regeneration that could have implications for hearing loss. Although it was felt

there could be more functional electrophysiology assessments we appreciate that this is beyond the current capability of the lab.

## Introduction

Hearing loss is a widespread public health issue affecting hundreds of millions of people worldwide. Hearing loss can be treated with cochlear implants or hearing aids but biological restoration of cochlear structure and function is not currently possible. Hearing is mediated by mechanosensitive hair cells in the organ of Corti, and loss or damage to these cells results in sensorineural hearing loss. Although mammals are only capable of very modest spontaneous hair cell regeneration in the balance organs (*Bramhall et al., 2014*; *Cox et al., 2014*; *Forge et al., 1993*; *Golub et al., 2012*; *Kawamoto et al., 2009*; *Kelley et al., 1995*; *Ogata et al., 1999*; *Rubel et al., 1995*), the cochlea lack this regenerative capability. This is in not the case in lower vertebrates. Robust turnover of hair cells is seen in the balance organs of many non-mammalian vertebrates (*Corwin, 1981*; *Jørgensen and Mathiesen, 1988*; *Lanford et al., 1996*; *Popper and Hoxter, 1990*). Impressive structural and functional recovery can also be achieved in the hearing organs of non-mammalian vertebrates following the killing of hair cells (*Adler and Raphael, 1996*; *Baird et al., 2000*; *Corwin and Cotanche, 1988*; *Cotanche, 1999*; *Niemiec et al., 1994*; *Raphael, 1993*; *Roberson et al., 2004*; *Roberson et al., 1996*; *Ryals and Rubel, 1988*). In these cases, supporting cells lying adjacent to hair cells can re-enter the cell cycle and trans-differentiate to generate new hair cells. These findings have prompted efforts to promote the regeneration of mammalian hair cells through genetic and pharmacological manipulations.

The basic helix-loop-helix transcription factor ATOH1 is both necessary and sufficient for hair cell development and survival (*Bermingham et al., 1999*; *Cai et al., 2013*; *Chonko et al., 2013*; *Driver et al., 2013*; *Pan et al., 2012*; *Woods et al., 2004*). In vitro studies using explants of the mammalian cochlea or inner ear balance organs showed that overexpression of ATOH1 can reprogram non-sensory cells of the inner ear into hair-cell-like cells (*Jen et al., 2019*; *Shou et al., 2003*; *Zheng and Gao, 2000*). Adenoviral expression of *Atoh1* in the cochlea of guinea pigs deafened with ototoxic drugs shows a variable and partial restoration of hearing after the lesion (*Izumikawa et al., 2005*). In vivo studies employing neonatal transgenic mice showed that cells of the greater epithelial ridge (GER) that lie next to the organ of Corti, and some supporting cells could be reprogrammed to hair-cell-like cells with the ectopic expression of *Atoh1* alone (*Kelly et al., 2012*; *Liu et al., 2012*). However, the hair cell reprogramming ability of ATOH1 declines rapidly with age (*Kelly et al., 2012*; *Liu et al., 2012*), suggesting a need for additional transcription factors to promote hair cell reprogramming in older animals. Moreover, cochleae where the auditory epithelium has degenerated further to state lacking hair cell and supporting cells (known as the flat epithelium; *Izumikawa et al., 2008*) do not respond to ATOH1 over-expression, further indicating the need for a more complex combinatorial approach.

Several transcription factors have been tested in combination with ATOH1 for their hair cell reprogramming potential (reviewed by *Iyer and Groves, 2021*). GFI1 and POU4F3 are two hair-cell-specific transcription factors expressed downstream of ATOH1 during development that has been implicated in hair cell survival and function (*Hertzano et al., 2004*; *Masuda et al., 2011*; *Wallis et al., 2003*; *Xiang et al., 1997*; *Xiang et al., 1998*). Adenoviral delivery of ATOH1 and GFI1 to adult mice in which hair cells were ablated promoted regeneration through supporting cell transdifferentiation at a higher efficiency than ATOH1 alone (*Lee et al., 2020*). Similarly, transgenic over-expression of combinations of ATOH1, GATA3, and POU4F3 reprogrammed adult supporting cells into hair-cell-like cells with improved efficiency (*Walters et al., 2017*). A combination of ATOH1, GFI1, and POU4F3 reprogrammed embryonic stem cells and chick otic epithelial cells in vitro to cells that expressed several hair cell genes, and showed key hair cell features (*Costa et al., 2015*). The co-overexpression of these three factors in vivo can also reprogram neonatal Lgr5 +supporting cells into hair-cell-like cells more efficiently than ATOH1 alone (*Chen et al., 2021*). Finally, the addition of SIX1 to the three factor cocktail was able to reprogram adult mouse tail-tip fibroblasts into hair-cell-like cells which have some epigenetic and transcriptional characteristics of hair cells, as well as transduction channel protein expression, and hair-cell-like electrophysiological properties (*Menendez et al., 2020*).

Recent studies have shown that one reason for the inability of cochlear supporting cells to convert to hair cells is that the chromatin surrounding hair cell genes becomes progressively less accessible

as the ear matures (*Jen et al., 2019*; *Tao et al., 2021*). The use of multiple hair cell transcription factors to reprogram supporting cells into hair cells may enhance the accessibility of hair cell loci in supporting cells, and recent evidence suggests that some hair cell transcription factors such as POU4F3 can do so in the developing cochlea by acting as pioneer factors (*Yu et al., 2021*). However, the question of whether combinations of multiple transcription factors simply improve the efficiency of hair cell reprogramming, or whether they can also improve the fidelity of hair cell reprogramming by activating a larger number of hair cell genes is currently unknown.

In this study, we sought to address this question by comparing the reprogramming potential of three transcription factor combinations – ATOH1 alone, ATOH1 + GFI1, and ATOH1 + GFI1+POU4F3 - in the mouse cochlea. We generated three transgenic mouse lines in which the transcription factor combinations were expressed from the ROSA26 locus in a Cre-dependent fashion. We found that ATOH1 alone is sufficient to reprogram neonatal non-sensory cells of the greater epithelial ridge into a mosaic of large numbers of hair cell-like cells that are surrounded by GLAST-positive supporting cell-like cells. The reprogrammed hair cells resembled inner hair cells and possessed stereocilia and some mechanotransduction properties. At these young ages, additional transcription factors do not enhance the number of new hair cells generated by ATOH1, nor do they increase the number of hair cells genes expressed in these reprogrammed cells, determined by single-cell RNA-seq. However, we show that after the first postnatal week, the overexpression of GFI1 and POU4F3 is necessary to enhance the hair cell reprogramming ability of ATOH1 in 8-day-old supporting cells. We also show that some supporting cell populations remain refractory to reprogramming even with three transcription factors, likely due to the action of the reprogramming factors being blocked by Notch signaling delivered by the endogenous hair cells. By simultaneously comparing the transcriptome and chromatin accessibility of cochlear cells at birth and 1 week of age using single-cell multi-omic approaches, we showed that hair cell loci become progressively less accessible in supporting cells and non-sensory cells of the cochlea during the first postnatal week. Our work provides the first mechanistic analysis of hair cell reprogramming and reveals some of the epigenetic and cell signaling obstacles that will need to be overcome in a therapeutic context in the mature inner ear.

## Results
### Hair cell transcription factors promote highly efficient reprogramming of non-sensory cochlear tissue into hair-cell-like cells in the neonatal mouse

To directly compare the efficiency of different transcription factor combinations in hair cell reprogramming, we targeted them to the *Rosa26* locus using a modified *Ai3* targeting vector containing a transcriptional stop cassette flanked by loxP sites (*Madisen et al., 2010*). We used three different hair cell transcription factor combinations: ATOH1 alone, GFI1 & ATOH1, and GFI1, ATOH1 & POU4F3 (*Figure 1A*). Individual transcription factor coding regions were separated by a GSG-T2A self-cleaving peptide sequence to ensure comparable transcription factor expression levels (*Tang et al., 2009*). We were able to achieve correct targeting efficiency to the *Rosa26* locus of approximately 80% by co-electroporating our targeting vectors with a plasmid expressing Cas9 and a sgRNA targeting the *Rosa26* locus between the two homology arms. This high efficiency allowed us to obtain correctly targeted ES cell clones with multiple constructs in single electroporation. We verified the expression of the transcription factor proteins by culturing the mouse ES cell lines used to generate founders for the three targeted mouse lines in the presence of membrane-permeable TAT-Cre protein, followed by western blotting (*Figure 1—figure supplement 1A*).

We targeted overexpression of the three-hair cell transcription factor combinations to the greater epithelial ridge (GER) and supporting cells of the neonatal mouse organ of Corti using *Sox9-CreER* transgenic mice (*Figure 1B*; *Kopp et al., 2011*). We confirmed GER and supporting cell-specific targeting with this mouse line by administering tamoxifen to one-day-old (P1) *Sox9-CreER; Rosa*<sup>EGFP</sup> (Ai3) mice and analyzing their cochleae a week later (P8; *Figure 1C*). The pattern of recombination in GER cells and apical turn supporting cells corresponded to the normal expression of SOX9 protein at this age, revealed by EGFP expression and the absence of recombination in hair cells (*Figure 1—figure supplement 1B*). All three *Rosa26*-targeted mouse lines were bred with *Sox9-CreER* to obtain experimental mice harboring both alleles. For the remainder of the manuscript, we will refer to mice

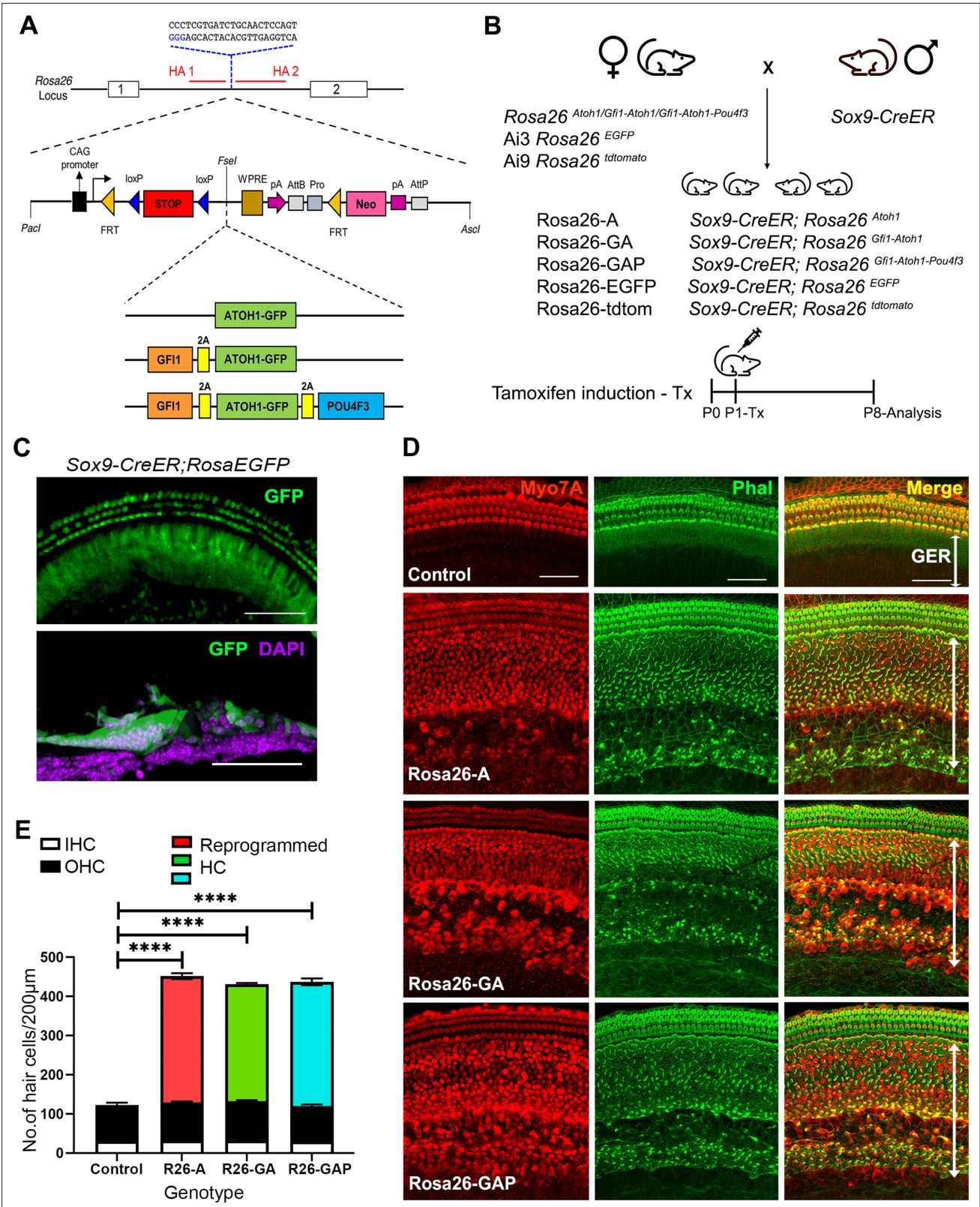

**Figure 1.** Non-sensory cells of the neonatal mouse cochlea can be efficiently reprogrammed to a hair cell fate with combinations of Atoh1, Gfi1 and Pou4f3 transcription factors. (**A**) Schematic representation of the strategy to target the Rosa26 locus to generate three conditional mouse lines for transcription factor overexpression. A modified Ai3 vector (***Madisen et al., 2010***) was used to target different transcription factor combinations to the *ROSA26* locus. ES cell targeting was enhanced using CRISPR-mediated cleavage with a sgRNA sequence targeting the *ROSA26* locus between the

*Figure 1 continued on next page*

*Figure 1 continued*

targeting homology arms (HA1 and 2). The transcription factor coding sequences were separated by GSG-T2A self-cleaving peptide sequences to generate multiple proteins from a single primary transcript. (**B**) Mating schemes to express different transcription factor combinations in the cochlea. The *Sox9-CreER* mouse was bred to the three Rosa26 overexpression lines and reporters to generate experimental animals of the following genotypes: Rosa26-A (*Sox9-CreER; Rosa^Atoh1GFP*), Rosa26-GA (*Sox9-CreER; Rosa^Gfi1-Atoh1GFP*), Rosa26-GAP (*Sox9-CreER; Rosa^Gfi1-Atoh1GFP-Pou4f3*), Rosa26^EGFP (*Sox9-CreER; Rosa26^EGFP*), and Rosa26-tdtomato (*Sox9-CreER; Rosa26^tdtomato*). Animals received tamoxifen (25 mg/kg body weight) at P1 and were sacrificed at P8. (**C**) GFP reporter expression obtained from mating *Sox9-CreER* mice with *Rosa26^EGFP* mice. Fluorescence is seen in all GER cells in whole mounts and 16 μm sections. Images show GFP (green) and a DAPI nuclear stain (magenta). Scale bar: 50 μm. (**D**) Large numbers of reprogrammed hair cells (white arrows) are seen in P8 cochleae extending from the organ of Corti to the interdental cell region in *Rosa-A, Rosa-GA,* and *Rosa-GAP* mice, revealed by immunostaining for Myosin VIIA (red) and Phalloidin (green). (**E**) Quantification of hair cells in the P8 reprogrammed cochleae. The number of Myosin VIIA + cells per 200 μm length of the cochlea was measured (IHC – Inner hair cells, OHC – Outer hair cells). Compared to controls, significant numbers of reprogrammed cells (300–320 per 200 μm) were seen in Rosa26-A, Rosa26-GA and Rosa26-GAP genotypes (n=3 per genotype). An unpaired t-test was performed to compare hair cell numbers between genotypes. The significant differences are represented. ****$P<0.00001$. Data are presented as mean ± SEM.

The online version of this article includes the following source data and figure supplement(s) for figure 1:

**Source data 1.** Overexpression of the ROSA-A, ROSA-GA, and ROSA-GAP transcription factor combinations from the Rosa26 locus was verified by culturing ES cells used to generate the three lines of mice with membrane soluble TAT-Cre.

**Source data 2.** Overexpression of the ROSA-A, ROSA-GA, and ROSA-GAP transcription factor combinations from the Rosa26 locus was verified by culturing ES cells used to generate the three lines of mice with membrane soluble TAT-Cre.

**Figure supplement 1.** Validation of transcription factor expression in ES cell lines used to generate *ROSA26*-targeted mice, and cochlear expression of the *Sox9-CreER* line.

**Figure supplement 2.** Reprogrammed hair cells at neonatal ages can survive until at least P15.

carrying the *Sox9-CreER* allele and the *Rosa26*-targeted transcription factor combinations as *Rosa-A*, *Rosa-GA*, and *Rosa-GAP* (*Figure 1B*). We activated each combination of transcription factors in the GER and supporting cells by injecting tamoxifen at P1 and analyzing mice one week later.

We immunostained the 8-day-old reprogrammed cochleae for the hair cell marker Myosin VIIA and the presence of actin-rich hair bundles with fluorescently-labeled phalloidin. We observed efficient reprogramming of GER cells into hair cell-like cells (*Figure 1D*), with large numbers of reprogrammed Myosin VIIA+/phalloidin + cells throughout the GER, extending from the neural edge of the organ of Corti to the interdental cell region (*Figure 1D*). These ectopic cells could survive in the GER until at least 15 days after birth (*Figure 1—figure supplement 2*). Reprogrammed hair cells were present in similar numbers throughout the basal-apical axis of the cochlea, with an average of 300 reprogrammed hair cells per 200 μm, compared to an average of 28 inner hair cells and 90 outer hair cells in a corresponding 200 μm length of the organ of Corti. We did not observe significant differences in reprogrammed hair cell numbers between the three transcription factor combinations at this age (*Figure 1E*). We characterized the P8 reprogrammed hair cell-like cells by immunostaining for known hair cell and supporting cell markers (*Figure 2A*). The reprogrammed cells in the GER expressed VGLUT3, a known vesicular transport protein expressed in inner hair cells (*Obholzer et al., 2008*; *Ruel et al., 2008*; *Figure 2A*). The reprogrammed cells did not express Prestin, a motor protein specific to outer hair cells which is necessary for their electromotility and their contribution to cochlear amplification and tuning (*Zheng et al., 2000*). The reprogrammed hair-cell-like cells in the GER received innervation from auditory afferents, labeled with the TuJ1 antibody to βIII-tubulin. The hair-cell-like cells also stained with antibodies to the CTBP2 transcription factor, which also recognizes Ribeye, a major component of ribbon synapses formed between afferent neurons and hair cells (*Sheets et al., 2011*).

To further characterize the hair-cell-like cells, we used scanning electron microscopy to assess the morphology of reprogrammed hair cell stereocilia and compared it to that of endogenous hair cells. Under all three reprogramming conditions, reprogrammed hair cells throughout the GER had stereocilia-like protrusions from their apical surfaces, possessing a staircase-like arrangement of hair bundles that appeared similar to control hair cells of the same age (*Figure 2B*). To determine the presence of mechanotransduction channel activity, we incubated explants of our P8 cochleae with the styryl dye FM 1–43, which permeates transduction channels. Hair cells mature in a basal-apical gradient along the cochlear duct, and between P6 and P7, all hair cells in the cochlea have matured to the point where they are permeable to FM1-43 dye (*Lelli et al., 2009*). Reprogrammed hair cells in the GER in all three conditions took up the FM 1–43 dye within 10 s (*Figure 2B*), although the degree

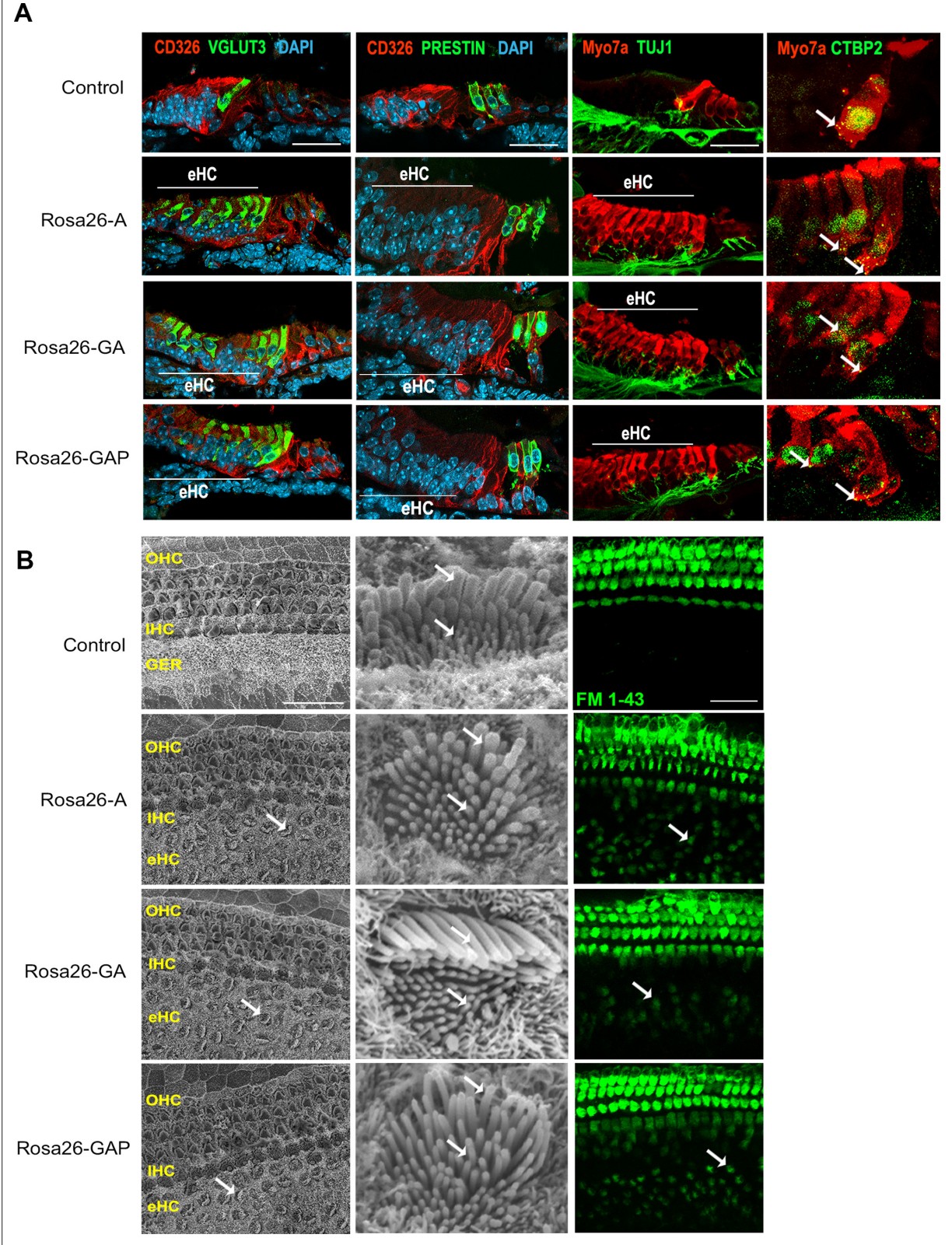

**Figure 2.** Day 8 reprogrammed hair cells are inner hair cell-like, innervated, form ribbon synapses, possess stereociliary bundles, and show evidence of mechanotransduction activity. (**A**) Control and reprogrammed cochleae were immunostained for an inner hair cell-specific marker, VGLUT3, an outer hair-cell-specific marker, PRESTIN, a GER specific marker, CD326/EpCAM, a neuronal marker TuJ1, a ribbon synapse-specific marker, and a hair cell marker, Myosin VIIA. The reprogrammed hair cell region is indicated (white line; eHC – ectopic hair cells). Arrows indicate individual ribbon synapse

*Figure 2 continued on next page*

*Figure 2 continued*

structures observed in the cell bodies of hair cells. Scale bar: 50 μm. (**B**) Scanning electron micrographs (SEM) of the control and reprogrammed cochleae (500 X; scale bar- 50 μm). OHC: Outer hair cell region, IHC: Inner hair cell region, GER: greater epithelia ridge region. Arrows indicate individual reprogrammed hair cells in the GER. SEM mages at 10,000 X show the arrangement of stereocilia in control and reprogrammed hair cells. Arrows indicate variations in the length of individual stereocilia which are similar between control and reprogrammed hair cells. The presence of mechanotransduction activity in the induced hair cells was tested by uptake of FM1-43 dye after 10 s of exposure. Reprogrammed hair cells in the GER take up the dye to a lesser extent than endogenous hair cells (arrows), but more than controls, indicating some mechanotransduction channel activity.

The online version of this article includes the following figure supplement(s) for figure 2:

**Figure supplement 1.** Transcription factor induction at P1 does not influence cell proliferation in the reprogrammed cochlea.

of labeling of the reprogrammed cells in the GER was significantly fainter than the endogenous hair cells visible in the organ of Corti. In sum, we established that under all three combinations of hair cell transcription factors, we generated large numbers of reprogrammed inner hair cell-like cells that are innervated, are morphologically similar to endogenous hair cells, show ribbon synapse formation, and possess some mechanotransduction channel activity.

To test if cell proliferation played a role in the generation of the reprogrammed hair cells in the GER, we assayed cell proliferation in the reprogrammed cochleae using EdU incorporation. Experimental and control animals were injected with tamoxifen at P1 to initiate transcription factor over-expression, followed by EdU injections (50 mg/kg body weight) twice every other day until P8. We observed that cell proliferation occurred only in the spiral ganglion region and not in the organ of Corti of experimental or control animals, and none of the reprogrammed hair cells in the GER were labeled by EdU (*Figure 2—figure supplement 1*). These data suggest that the vast majority of the reprogrammed hair cells we observe in the GER are generated by direct non-mitotic reprogramming, rather than by proliferation.

## scRNA-seq analysis reveals that cochlear reprogramming in newborn mice generates new hair cells that are similar to their endogenous counterparts

Our data suggested that the gross phenotype of the reprogrammed hair cell-like in neonatal mice resembled wild type hair cells, and did not vary significantly between the three reprogramming conditions. To determine whether unique transcriptional changes occurred in response to the three reprogramming conditions, we performed single-cell RNA-sequencing analysis of the reprogrammed cells. We bred the *Sox9-CreER* mice to the Ai9 *Rosa-tdTomato* reporter line and further bred these to the three Rosa26 conditional overexpression lines (*Figure 3A*) to obtain our experimental mice, where one ROSA26 allele carried a tdTomato reporter, and the other ROSA26 allele carried one of the three reprogramming cassettes. These mice were injected with tamoxifen at P1 and tdTomato-positive cochlear cells were purified by FACS at P8 (*Figure 3A*) and used to generate scRNA-seq libraries using the 10xGenomics Chromium platform.

The cell clustering pattern observed after the integration of cells from all four genotypes allowed us to identify expected cell-type-specific clusters based on transcriptomic data from previous studies (*Kolla et al., 2020*). Sensory and non-sensory cells of the cochlear duct, including hair cells, supporting cells, greater epithelial ridge cells, cells of the stria vascularis, and glial cells were all identified in the clustering (*Figure 3B*). Examples of marker genes used to validate cluster identification on the basis of their expression in the hair cell and lateral GER clusters is shown in *Figure 3—figure supplement 1A*. Consistent with our successful FM1-43 labeling of the new hair cells, we observed significant expression of genes associated with hair cell mechanotransduction (*Tmc1, Tmie, Lhfpl5, Cdh23, Pcdh15*) compared to GER or supporting cells (*Figure 3—figure supplement 1B*). Our clustering analysis confirmed the results obtained by staining with cell-type-specific markers: we saw a reduction in cells of the GER (particularly lateral GER) but a significant increase in the numbers of hair cells in the three reprogrammed conditions compared to control mice (*Figure 3C*). Other cochlear cell types that were identified during this analysis are indicated in the diagram in *Figure 3D*. We performed a gene ontology analysis (GO- Biological process) to ascertain the morphological and functional characteristics of these reprogrammed hair cells. We identified 496 genes significantly expressed genes in the reprogrammed hair cells across all three overexpression conditions (cut off p-value $< 1.00E^{-25}$) which

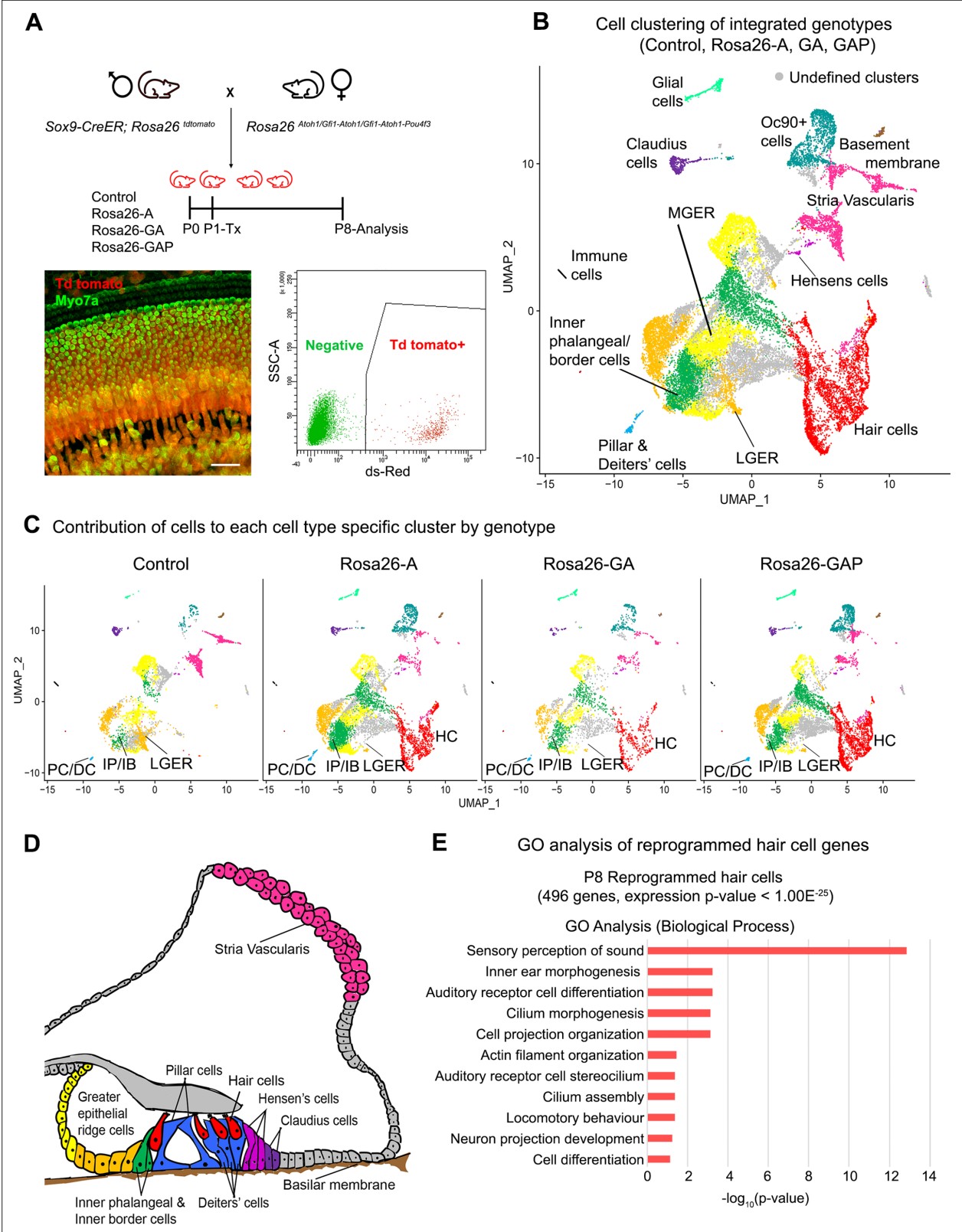

**Figure 3.** Single-cell transcriptomic analysis of control and reprogrammed P8 GER cells confirm the presence of a large number of reprogrammed cells that possess hair cell-like gene signatures. (**A**) Mice carrying a *Sox9-CreER* allele, a *ROSA26^tdTomato* reporter allele, and a modified ROSA26 allele containing reprogramming factors received tamoxifen at day 1 and tdTomato + cells were purified by FACS sorting one week later. A representative whole-mount image of a day 8 cochlea shows reprogrammed hair cells and the tdTomato reporter (scale bar: 50 μm). A representative FACS plot of

*Figure 3 continued on next page*

*Figure 3 continued*

dissociated cochlear cells is shown. tdTomato + cells were collected and analyzed by scRNA-seq. (**B**) UMAP plot for cells integrated and analyzed from all four genotypes (Control, Rosa26-A, Rosa26-GA, and Rosa26-GAP) purified in (**A**). Each identified cluster has been labeled and the anatomical location of each cluster is shown color-coded in panel (**D**). (**C**) Genotype-wise UMAP plots highlighting the contribution of cells from each genotype in every identified cluster. The GER cluster (particularly LGER) in the control is prominent and the hair cell cluster is present only in the reprogrammed cochlear genotypes as the *Sox9-CreER* line does not label endogenous hair cells. IP/IB – Inner phalangeal/border, PC/DC – Pillar/Deiters' cells, HC – Hair cells, LGER – Lateral Greater epithelial ridge (**D**) Schematic of an organ of Corti cross-section at P8. Unique cell types identified in the scRNA-seq clustering have been color-coded and correspond to the cluster colors in (**B**) and (**C**). (**E**) Gene ontology analysis of the top differentially expressed genes in the reprogrammed hair cell-like cells from all three conditions (with respect to their expression in other cell clusters). A list of ~500 significantly expressed genes ($P<1.00E^{-25}$) was analyzed and GO terms (Biological process; $-\log_{10}$ (p-value)>1) are shown.

The online version of this article includes the following figure supplement(s) for figure 3:

**Figure supplement 1.** Examples of hair cell and LGER marker genes confirm cluster assignments in P8 cochlear cell clusters.

were used as input for gene ontology analysis. The top GO hits included genes for sensory perception of sound (GO: 0007605, 27 genes, $p=1.40E^{-13}$), inner ear morphogenesis (GO: 0042472, 12 genes, $p=5.70E^{-04}$), auditory receptor cell differentiation (GO: 0042491, 7 genes, $p=5.70E^{-04}$), cilium morphogenesis (GO:0060271,17 genes, $p=7.50E^{-04}$) and cell projection organization (GO:0030030,16 genes, $p=7.50E^{-04}$; *Figure 3E*). Together, our data suggest that reprogrammed hair cells possess many morphological and transcriptional characteristics similar to endogenous inner hair cells. However, they also show that at this age, the addition of Gfi1 and Pou4f3 does not improve the morphology of the reprogrammed hair cells, nor a more complete complement of hair cell genes expressed in the reprogrammed cells.

## Overexpression of GFI1 and POU4F3 enhances the hair cell reprogramming ability of ATOH1 in the mouse cochlea at older ages

Our data show that all three transcription factor combinations have similar hair cell reprogramming potential in the neonatal mouse cochlea. Previous overexpression studies have shown that the reprogramming efficiency of ATOH1 declines with increasing age (*Kelly et al., 2012*; *Liu et al., 2012*). We next explored the in vivo reprogramming potential of our transcription factor combinations in older animals. Employing the same experimental mouse breeding scheme as described above, we overexpressed the three transcription factor combinations (ATOH1, GFI1 & ATOH1 and GFI1, ATOH1 & POU4F3) in the GER and organ of Corti, including all supporting cells, again using *Sox9-CreER* mice (*Figure 4A*). We confirmed correct and efficient recombination in GER and supporting cells by administering tamoxifen in one week old (P8) *Sox9-CreER; Rosa*$^{EGFP}$ mice and analyzing GFP reporter expression by immunostaining a week later (P15; *Figure 4B*). Between P7 and P15, parts of the GER undergo remodeling through thyroid-hormone-dependent apoptosis and are replaced by cuboidal inner sulcus cells (*Peeters et al., 2015*). By P15 several rows of SOX9 +GER cells remain on the lateral edge of the inner sulcus adjacent to the organ of Corti. These are still targeted correctly by *Sox9-CreER* mice (*Figure 4—figure supplement 1A*).

We compared overexpression of the three transcription factor combinations by giving tamoxifen at P8 to *Rosa-A, Rosa-GA,* and *Rosa-GAP* mice harboring the *Sox9-CreER* allele to target the GER and supporting cells. Analysis of the mice a week later (P15) with the hair cell markers Myosin VIIA and phalloidin revealed that reprogrammed hair cells were significantly higher in the *Rosa-GAP* condition (average GAP reprogrammed hair cells – 55, Inner hair cells- 31, Outer hair cells – 95 per 200 μm length of the organ of Corti; *Figure 4D*). By using an Ai9 *Rosa-tdTomato* reporter line, we demonstrated that these new hair cells were derived from Sox9-expressing supporting cells or GER cells (Figure 6A). These new hair cells continued to survive until at least P29 and showed increasing organization of phalloidin-stained hair bundles with increasing age (*Figure 4—figure supplement 1B*). Immunostaining of the reprogrammed cells revealed that, unlike the younger reprogrammed cells, P15 reprogrammed hair cells in the Rosa26-GAP condition did not express the inner hair cell marker VGLUT3 (*Figure 5A*). These reprogrammed hair cells did, however, show evidence of innervation based on staining with TuJ1 antibody, and formed ribbon synapses based on positive staining for CTBP2 (*Figure 5A*). We used scanning electron microscopy (SEM) to assess the morphology of reprogrammed hair cell stereocilia across all three conditions. Low power images showed the presence of sparse reprogrammed hair cells in the ATOH1 and GFI1+ATOH1 overexpression conditions which

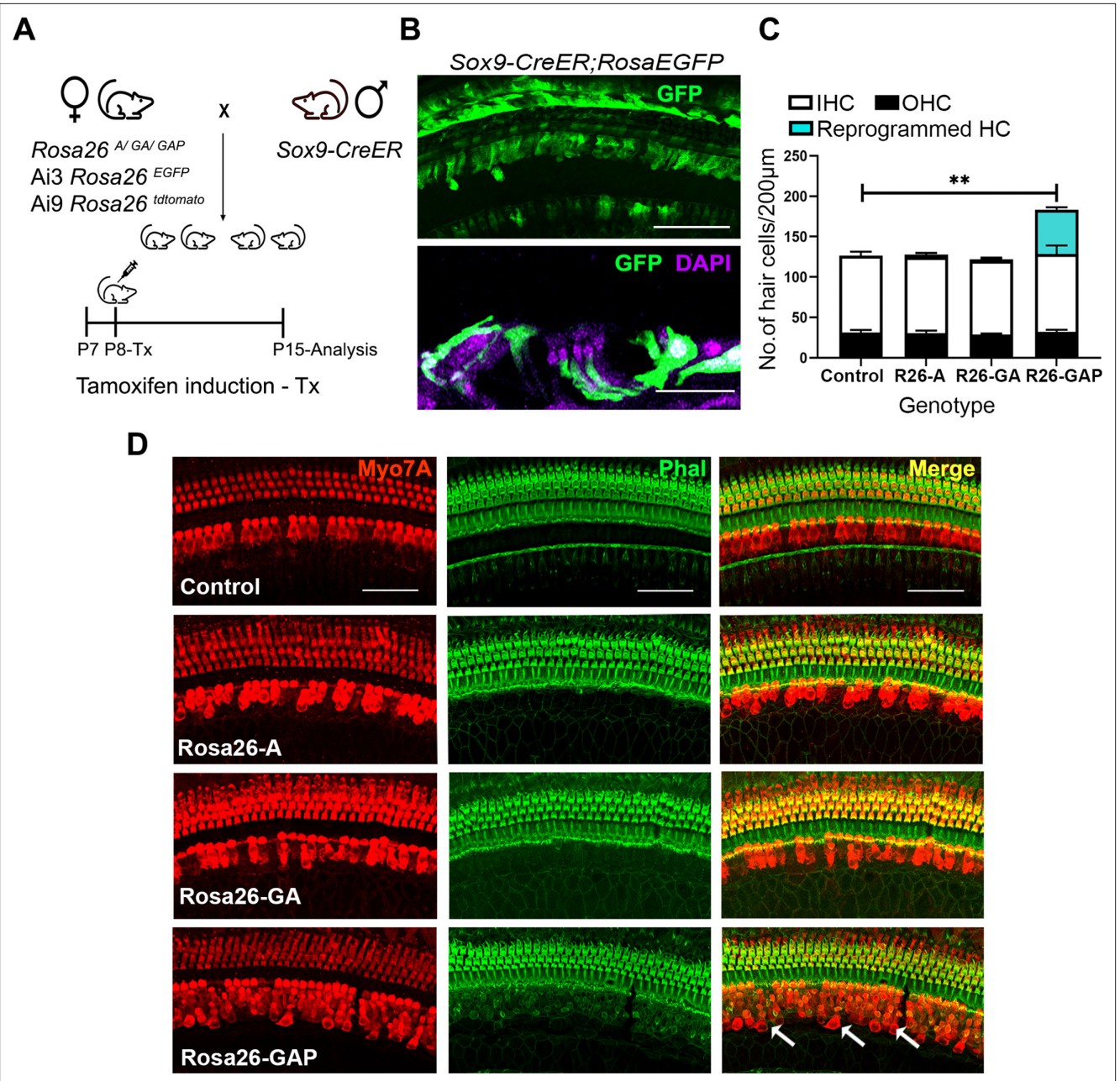

**Figure 4.** *Gfi1, Atoh1, Pou4f3,* but not *Atoh1* or *Gfi1+Atoh1,* can reprogram 8-day-old GER cells into hair cell-like cells. (**A**) Mating scheme for the targeting of transcription factors to the greater epithelial ridge and all supporting cells. The *Sox9-CreER* mouse is bred to the three Rosa26 overexpression lines in a similar manner to *Figure 1A*. Animals received tamoxifen (25 mg/kg body weight) at P8 and were sacrificed at P15. (**B**) GFP reporter expression in some lateral GER cells and all supporting cells detected by immunostaining in the organ of Corti of the *Sox9-CreER; Rosa26*^EGFP cochlea. Images show detection of GFP (green) and nuclear stain, DAPI (magenta). Scale bar: 50 μm. (**C**) Quantification of hair cells in the P15 reprogrammed cochleae. The number of Myosin VIIA + cells per 200 μm length of the organs of Corti from control, Rosa26-A, Rosa26-GA and Rosa26-GAP genotypes (n=3 per genotype) are represented. *Rosa-GAP* mice show approximately 50–60 ectopic hair cells, whereas *Rosa-A* and *Rosa-GA* show less than 5 ectopic cells per 200 μm. An unpaired t-test was performed to compare hair cell numbers between genotypes. Significant differences are represented. **p<0.001. Data are presented as mean ± SEM. (**D**) Rosa26-GAP mice can reprogram GER cells to hair cell-like cells. Immunostaining for Myosin VIIA (red) and Phalloidin (green) in the P15 cochleae (whole-mount organ of Corti - 200 μm length) of control, *Rosa26-A, Rosa26-GA,* and *Rosa26-GAP* mice. Arrows point to the GER region in the *Rosa26-GAP* cochlea, where many reprogrammed hair cells are observed.

The online version of this article includes the following figure supplement(s) for figure 4:

**Figure supplement 1.** Validation of *Sox9-CreER* activity in the cochlea from P8 to P15 and survival of reprogrammed hair cells from P15 to P29.

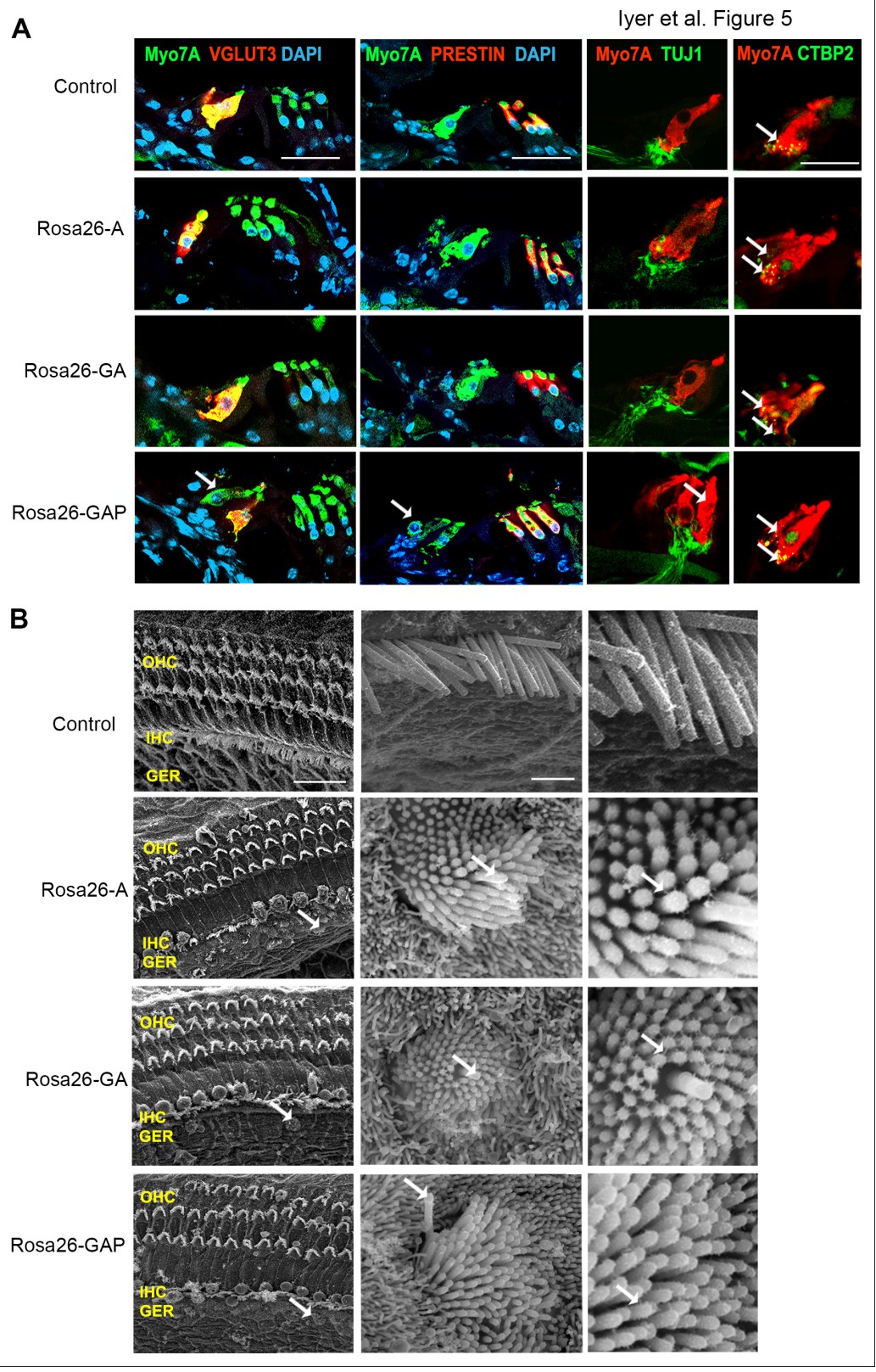

Iyer et al. Figure 5

**Figure 5.** Postnatal (**P15**) Rosa26-GAP reprogrammed hair cells are innervated and form ribbon synapses, but possess immature stereociliary bundles. (**A**) Control and reprogrammed cochleae immunostained for the inner hair cell-specific marker, VGLUT3, outer hair-cell-specific marker, PRESTIN, neuronal marker TuJ1, ribbon synapse-specific marker CTBP2 and hair cell marker, Myosin VIIA. Arrows point to reprogrammed hair cells that are positive

*Figure 5 continued on next page*

*Figure 5 continued*

for Myosin VIIA in the *Rosa26-GAP* condition, innervation of the reprogrammed hair cells, and individual ribbon synapse structures observed in the cell bodies of endogenous and reprogrammed hair cells (*Rosa-GAP*). Images show detection of described markers on a 16 µm section of the organ of Corti (control and reprogrammed). Scale bar: 50 µm. (**B**) Scanning electron micrographs of reprogrammed hair cells from all three genotypes show similar hair cell-like structural features. Scanning electron micrographs (SEM) of the control and reprogrammed cochleae at 1000 X (scale bar- 50 µm). Arrows indicate individual reprogrammed hair cells. OHC: Outer hair cell region, IHC: Inner hair cell region, GER: greater epithelia ridge region. SEM images at 50,000 X show the kinocilium on individual hair cells and side link structures connecting hair cell stereocilia as indicated by arrows.

The online version of this article includes the following figure supplement(s) for figure 5:

**Figure supplement 1.** FM1-43X labeling of P15 reprogrammed hair cells.

did not stain for any hair cell markers other than Myosin VIIA (*Figure 5B*). Higher magnification SEM images at 50,000 X revealed that individual hair cell stereocilia of reprogrammed hair cells in all three conditions were immature compared to endogenous hair cells and the P1 reprogrammed hair cells we observed at P8. Many of the ectopic cells possessed kinocilia but the stereocilia did not exhibit a staircase-like structure and still contained many side links between individual stereocilia, indicating their immature state (*Figure 5B*). Some of the ectopic hair cells produced in all three conditions were able to take up the dye FM1-43X, suggesting that at least some of the cells have some rudimentary mechanotransduction activity (*Figure 5—figure supplement 1*). Together, we established that all three overexpression conditions are capable of producing reprogrammed hair-cell-like cells at P15, but the Rosa26-GAP cocktail is significantly more efficient at producing new hair cells. Nevertheless, these reprogrammed hair cells created between P8 and P15 are less mature than those derived from reprogramming neonatal cells between P1 and P8.

To determine whether the hair cells generated by overexpression of ATOH1, GFI1, and POU4F3 expressed more elements of the hair cell gene regulatory network, we repeated the scRNA-seq analysis described above on our three mouse lines, applying tamoxifen at day 8 and sorting and analyzing cells at P15 (*Figure 6A*). The genotype-integrated cell clustering pattern obtained allowed us to identify expected cell-type-specific clusters based on marker expression data from prior studies (*Ranum et al., 2019*). We identified multiple cell types in the clustering, including glial cells, hair cells, supporting cells, cells of the stria vascularis, spiral limbus, and interdental cells (*Figure 6B*). Examples of marker genes used to validate cluster identification on the basis of their expression in the hair cell and supporting cell clusters is shown in *Figure 6—figure supplement 1A*. Initial clustering analysis confirmed our earlier findings that the number of reprogrammed hair cells obtained in response to overexpression of GAP factors is greater than the small number of hair cells seen with GFI1 +ATOH1 or ATOH1 alone (*Figure 6C*). Other cochlear cell types identified in the analysis are indicated in the organ of Corti diagram using identical color coding to the UMAP plots (*Figure 6D*).

To elucidate the characteristics of the reprogrammed hair cells, we identified a list of 200 significantly expressed genes in hair cells obtained from each of the three overexpression conditions (cut off p-value < 1.00E$^{-15}$). We performed a gene ontology analysis to ascertain the overall characteristics of these reprogrammed hair cells and look for possible differences (*Figure 6E*). The top GO terms included genes for cell projection (GO:0042995), cytoskeleton (GO:0005856), cilium (GO:0005929) - Cellular component, Cell projection organization (GO: 0030030) - Biological process, calcium ion binding (GO: 0005509), calmodulin-binding (GO: 0005516) - Molecular function. The reprogrammed hair cells also expressed some genes coding for proteins of the mechanotransduction apparatus (*Tmie, Lhfpl5, Pcdh15; Figure 6—figure supplement 1B*), although the range of mechanotransduction genes was less than in reprogrammed hair cells in P8 mice (*Figure 3—figure supplement 1B*). We were not able to detect significant levels of either the *Tmc1* or *Tmc2* channels at this age, consistent with the low amount of FM1-43X labeling in the ectopic hair cells. Taken together, we have shown that hair cells obtained from all three overexpression conditions are transcriptionally similar and possess immature hair cell-like features. However, although the *Rosa26-GAP* reprogramming mice generate more hair cells, their transcriptional profile did not differ significantly from hair cells observed in *Rosa-A* or *Rosa-GA* conditions, suggesting that the additional reprogramming factors increase the efficiency, but not the fidelity of hair cell reprogramming.

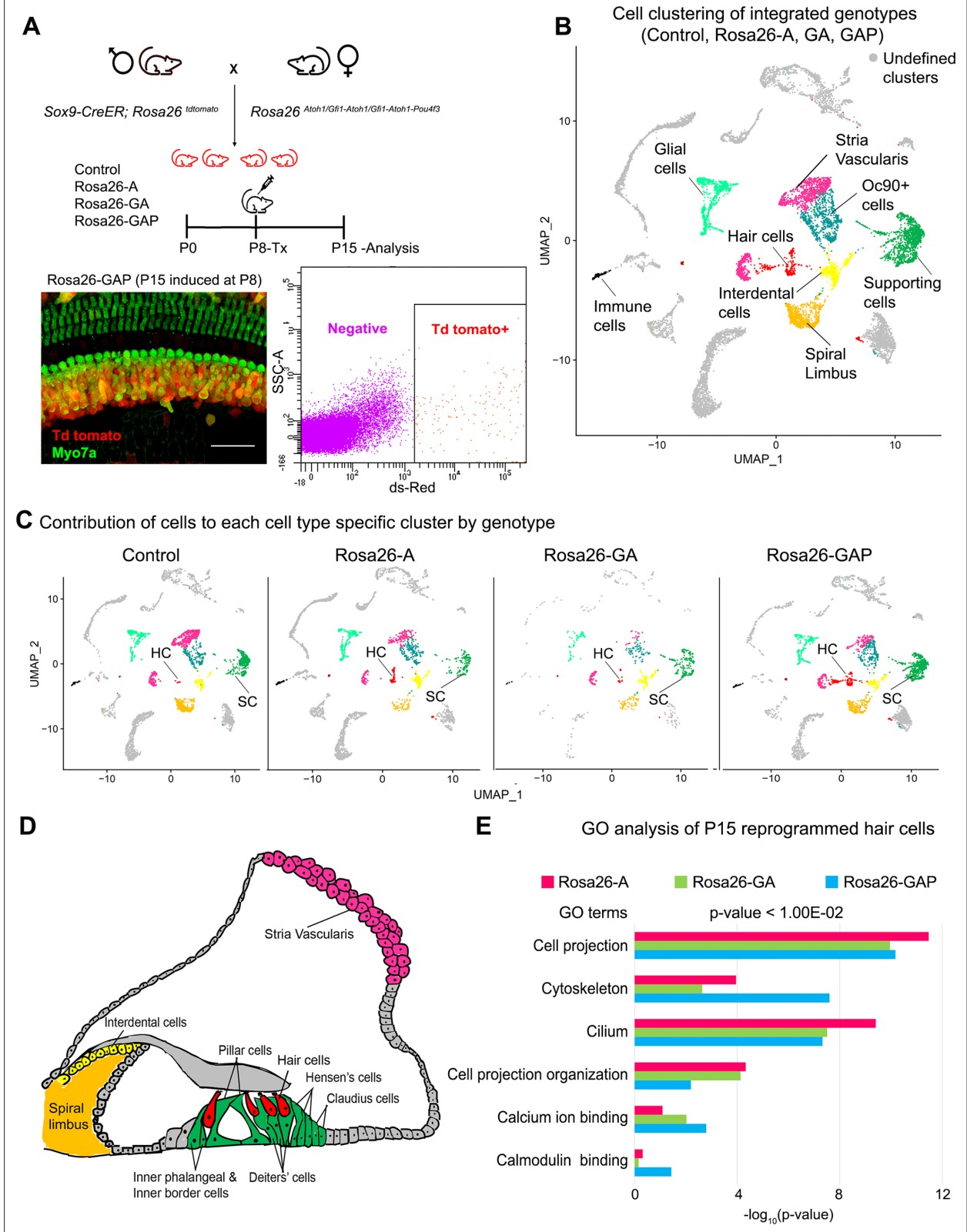

**Figure 6.** Single-cell transcriptomic analysis of control and reprogrammed cochlear cells at P15. (**A**) FACS-based enrichment of cochlear cells targeted for transcription factor overexpression. The breeding scheme with an experimental timeline is described followed by a representative whole mount image (bar: 50 μm) from the Rosa26-GAP cochlea. The scheme is similar to that shown in **Figure 3**, except that tamoxifen is injected to induce reprogramming at 8 days after birth, followed by analysis at day 15. All cells targeted for TF overexpression are tdTomato positive (red), including

*Figure 6 continued on next page*

Figure 6 continued

reprogrammed hair cells (green). A representative FACS scatter plot of dissociated induced cochlear cells is shown. (**B**) UMAP plot for cells integrated and analyzed by scRNA-seq from all four genotypes (control, Rosa26-A, Rosa26-GA, and Rosa26-GAP) purified in (**A**). Each identified cluster has been labeled. (**C**) Genotype-wise UMAP plots highlighting the contribution of cells from each genotype in every identified cluster. (**D**) Schematic of the organ of Corti cross-section at P15. Unique cell types have been color-coded and correspond to cluster colors in (**B**) and (**C**). (**E**) Gene ontology analysis of the top differentially expressed genes in reprogrammed hair cells from each condition (with respect to their expression in other cell clusters for that genotype). A list of ~200 significantly expressed genes (p<1.00E$^{-02}$) was analyzed and GO terms (Biological process, Cellular component, Molecular function; -log$_{10}$ (p-value)>1) are represented.

The online version of this article includes the following figure supplement(s) for figure 6:

**Figure supplement 1.** Examples of hair cell and supporting cell marker genes confirm cluster assignments in P15 cochlear cell clusters.

## Hair cell reprogramming of the greater epithelial ridge generates a mosaic of hair cells and supporting cells through activation of Notch signaling

A consistent observation in our reprogramming experiments conducted between P1-P8 and P8-P15 was that some supporting cells of the organ of Corti – the Deiters' cells and pillar cells – did not respond to the reprogramming factors by expressing hair cell proteins such as Myosin VIIA and did not exhibit any morphological changes indicating they were transforming into hair cells. We confirmed these results - obtained with *Sox9-CreER* mice - using a second Cre line, the *Lfng-CreER* line that causes efficient recombination in all supporting cell types in the organ of Corti (*Basch et al., 2016b*). Under all three reprogramming conditions at both stages (P1-P8 and P8-P15), we consistently failed to see conversion of Deiters' cells or pillar cells into Myosin VIIA + hair cells (*Figure 7—figure supplement 1*).

Signals from hair cells, particularly the Notch signaling pathway, are known to promote and stabilize supporting cell fate during development (*Basch et al., 2016b*; *Woods et al., 2004*) and induce supporting cell fate in the presence of ectopic hair cells (*Kelly et al., 2012*; *Woods et al., 2004*). Two types of supporting cells lie adjacent to inner hair cells: inner phalangeal and inner border cells. Both supporting cell types express the GLAST glutamate-aspartate-transporter, which plays a role in the uptake of neurotransmitters by inner hair cell ribbon synapses (*Glowatzki et al., 2006*). PROX1 is a marker unique to pillar and Deiters' cells of the outer hair cell region, which is expressed until the second week of age (*Bermingham-McDonogh et al., 2006*). Finally, all supporting cell types in the neonatal cochlea express the transcription factor SOX2. To determine if reprogrammed inner hair cell-like cells could promote the formation of supporting cell types normally associated with inner hair cells, we immunostained cochleae reprogrammed from P1-P8 for the supporting cell markers GLAST, PROX1, and SOX2. We found that cells in the reprogrammed GER lying beneath the reprogrammed Myosin VIIA+hair cells expressed GLAST and SOX2 protein, but not PROX1 (*Figure 7A*). This suggested the reprogrammed hair cells were able to promote the formation of inner phalangeal cell and border cell-like cells from the GER, even though these cells were also expressing the reprogramming factor combinations. We observed approximately equal numbers of supporting cells in the presence of all three transcription factor combinations, suggesting that even the presence of ATOH1, GFI1, and POU4F3 in GER cells was not sufficient to prevent them from forming supporting cell-like cells when apposed to reprogrammed hair cells.

We analyzed our single-cell RNA sequencing data to identify differences in the type of GLAST + supporting cell-like cells obtained in each condition by comparing them to wild-type (WT) inner phalangeal/border cells. Our genotype-based cell clustering data showed a significant increase in the inner phalangeal and border cell clusters between control and induced conditions (*Figure 3C*). A differential gene expression analysis for supporting cells in Rosa26-A vs WT, Rosa26-GA vs WT, and Rosa26-GAP vs WT indicated a common pattern of up- and downregulated genes (*Figure 7B*). *Cryab*, *Ccnd1*, *Rcn1*, and *Hes5* were upregulated in all three cases. *Cryab*, is a known heat shock protein with otoprotective effects during stress response and *Ccnd1*, is a cell cycle gene that is downregulated with the increasing maturity of supporting cells (*Erni et al., 2019*; *Laine et al., 2010*; *Sadler et al., 2020*). *Hes5* is a Notch-responsive gene that is expressed in Deiters' cells and pillar cells but not inner phalangeal and border cells at birth (*Doetzlhofer et al., 2009*; *Tateya et al., 2011*). Its expression in the reprogrammed GLAST positive cells is likely a response to active Notch signaling

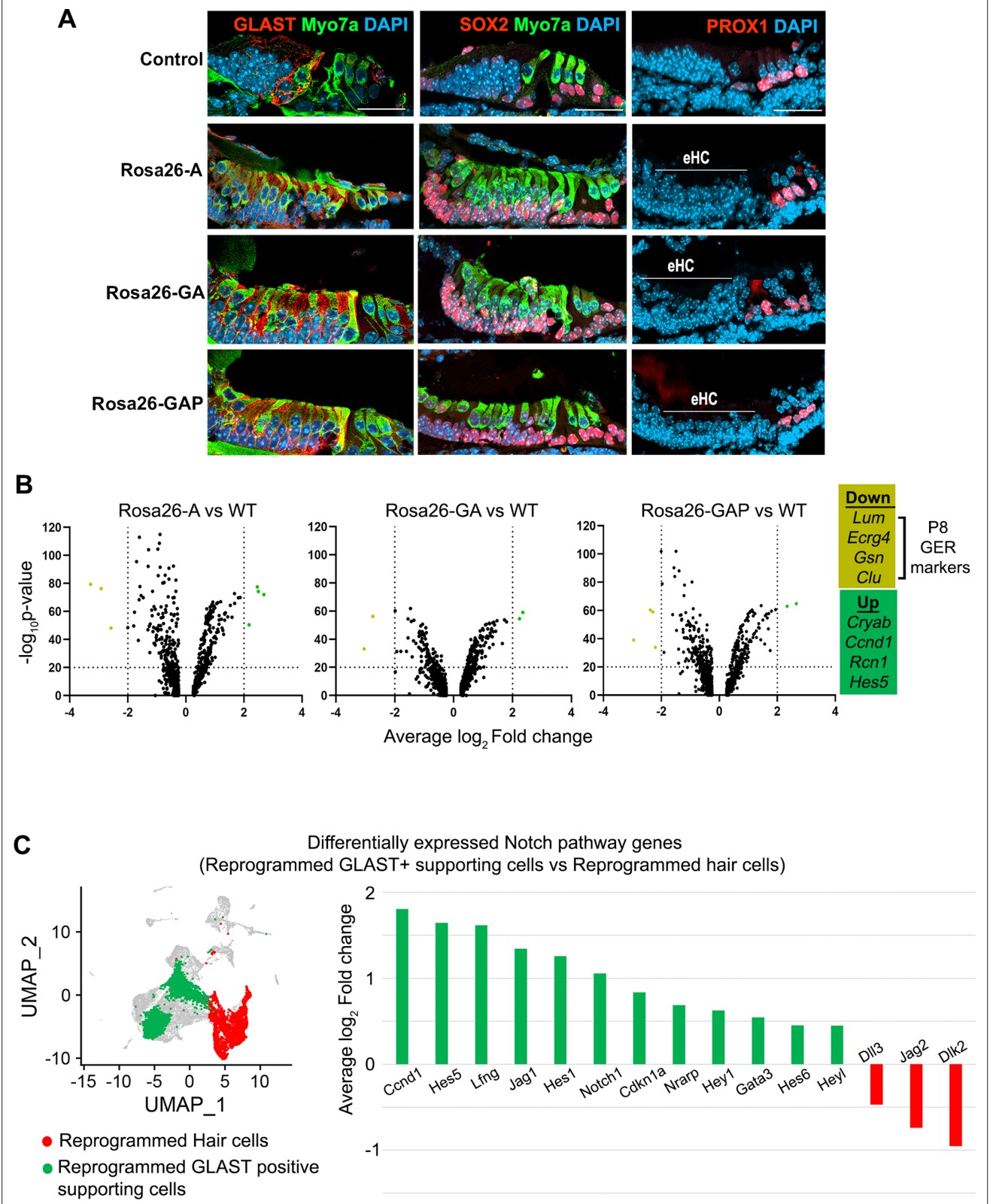

**Figure 7.** GLAST+, SOX2+supporting cells are induced adjacent to reprogrammed hair cells in the GER. (**A**) Control and reprogrammed cochleae immunostained for markers specific to inner phalangeal and border cells (GLAST), a general supporting cell marker (SOX2), pillar and Deiters' cells (PROX1), and the hair cell marker, Myosin VIIA. The reprogrammed hair cell region is indicated (line; eHC – ectopic hair cells). Images show 16 μm sections of the organ of Corti (control and reprogrammed). Scale bar: 50 μm. (**B**) Differentially expressed genes from our P1-P8 scRNA-seq experiments

*Figure 7 continued on next page*

*Figure 7 continued*

in reprogrammed GLAST + supporting cells are compared to control inner phalangeal/border cells. Volcano plots show common upregulated genes *Cryab, Ccnd1, Rcn1, Hes5,* and downregulated genes *Lum, Ecrg4, Gsn, Clu* (GER specific genes). (**C**) Notch pathway genes are upregulated in the reprogrammed GLAST + cells and hair cells in response to transcription factor induction at P8. UMAP plot of cells integrated from all genotypes is shown with the reprogrammed hair cells (red) and GLAST positive supporting cells (green). Average log$_{10}$ fold change in the expression of supporting cell-specific Notch genes – *Ccnd1, Hes5, Lfng, Jag1, Hes1, Notch1, Cdkn1a, Nrarp, Hey1, Gata3, Hes6, Heyl* and hair-cell-specific Notch genes – *Dll3, Jag2, Dlk2* is represented.

The online version of this article includes the following figure supplement(s) for figure 7:

**Figure supplement 1.** Use of the a second Cre line, *Lfng-CreER*, confirms that pillar and Deiters' cells do not get reprogrammed into hair cell-like cells in response to transcription factor overexpression.

---

induced by the ectopic hair cells to maintain supporting cell identity (*Wang et al., 2010*). We next examined known Notch pathway genes by performing a differential gene expression analysis between the reprogrammed GLAST + supporting cells and reprogrammed hair cells at P8. We observed the upregulation of Notch-receiving genes (*Lfng, Notch1,* and *Hes1*) in the reprogrammed supporting cells and hair cell-specific (*Dll3, Jag2,* and *Dlk2*) Notch ligand genes in the reprogrammed hair cells (*Figure 7C*). This suggests that transcription factor reprogramming is capable of reconstituting the Notch signaling interactions between hair cells and supporting cells, and that these interactions are sufficient to repress the action of the three reprogramming transcription factors in the ectopically induced supporting cells.

We repeated our supporting cell experiments by activating reprogramming at P8 and analyzing at P15. We saw evidence for the presence of ectopic GLAST+, SOX2+supporting cell-like cells adjacent to reprogrammed hair cells in the Rosa26-GAP condition alone (*Figure 8A*). EdU injections given every second day from P8 to P15 showed that none of the reprogrammed hair cells or supporting cells were generated by proliferation. (*Figure 8—figure supplement 1*). We examined our scRNA-seq data to determine the degree to which supporting cells alter their transcriptomes in response to the three different reprogramming combinations. We performed a differential gene expression analysis of all P15 supporting cells compared to their wild-type counterparts to analyze transcriptomic changes in response to each of the three transcription factor cocktails. The most significant up and down-regulated genes are highlighted (*Figure 8B*), and include Notch pathway-associated genes like *Mfng, Ccnd1, Hes5,* and *Dlk2*. In parallel to this, we also observed downregulation of many supporting cell genes such as *Ttll3, Rorb, Scd1, Scnn1b, Hhatl, Washc2* in addition to *Caecam16*. A complete description of these differentially expressed gene functions and their cell type-specific expression – extracted from the gEAR database (https://www.umgear.org/; *Orvis et al., 2021*) is given in *Figure 8*, *Figure 8—source data 1* and *Figure 8—source data 2*.

## Multi-omic analysis of the cochlea reveals hair cell loci become less epigenetically accessible in supporting cells and GER cells between postnatal days 1 and 8

The data described above suggest that both GER and supporting cells of the cochlea become more resistant to transcription factor reprogramming into hair cells during the first postnatal week. To determine if changes in the epigenetic accessibility of hair cell gene loci was partly responsible for this change, we used scMultiome to simultaneously profile gene expression and chromatin accessibility at the single cell level for each cell type of the cochlea in wild type day 1 and day 8 mice. We were able to identify most hair cell and supporting cell types of the organ of Corti as well as cells of the GER by clustering based on scRNA-seq, scATAC-seq and using 'weighted-nearest neighbor' analysis (WNN; *Hao et al., 2021*) which gave the clearest separation of cochlear cell types (*Figure 9A*). We extracted ATAC-seq profiles from 1627 distal regulatory elements associated with hair cell genes and generated heat maps to show the accessibility of these elements in GER cells and inner phalangeal cells and border cells in day 1 and day 8 cochlear tissue (*Figure 9B*). 498 elements showed comparable accessibility in these cell populations at both ages. However, 972 elements were significantly more accessible in the three cell populations at day 1 compared to day 8. A small number of distal elements (157) appeared to be somewhat more accessible in GER cells at day 8 compared to day 1. Examples of traces from some hair cell loci are shown in *Figure 9C*. *Hes6, Myo3b,* and *Pou4f3* all showed reduced accessibility at day 8 in distal or intergenic regulatory elements at older ages. These results

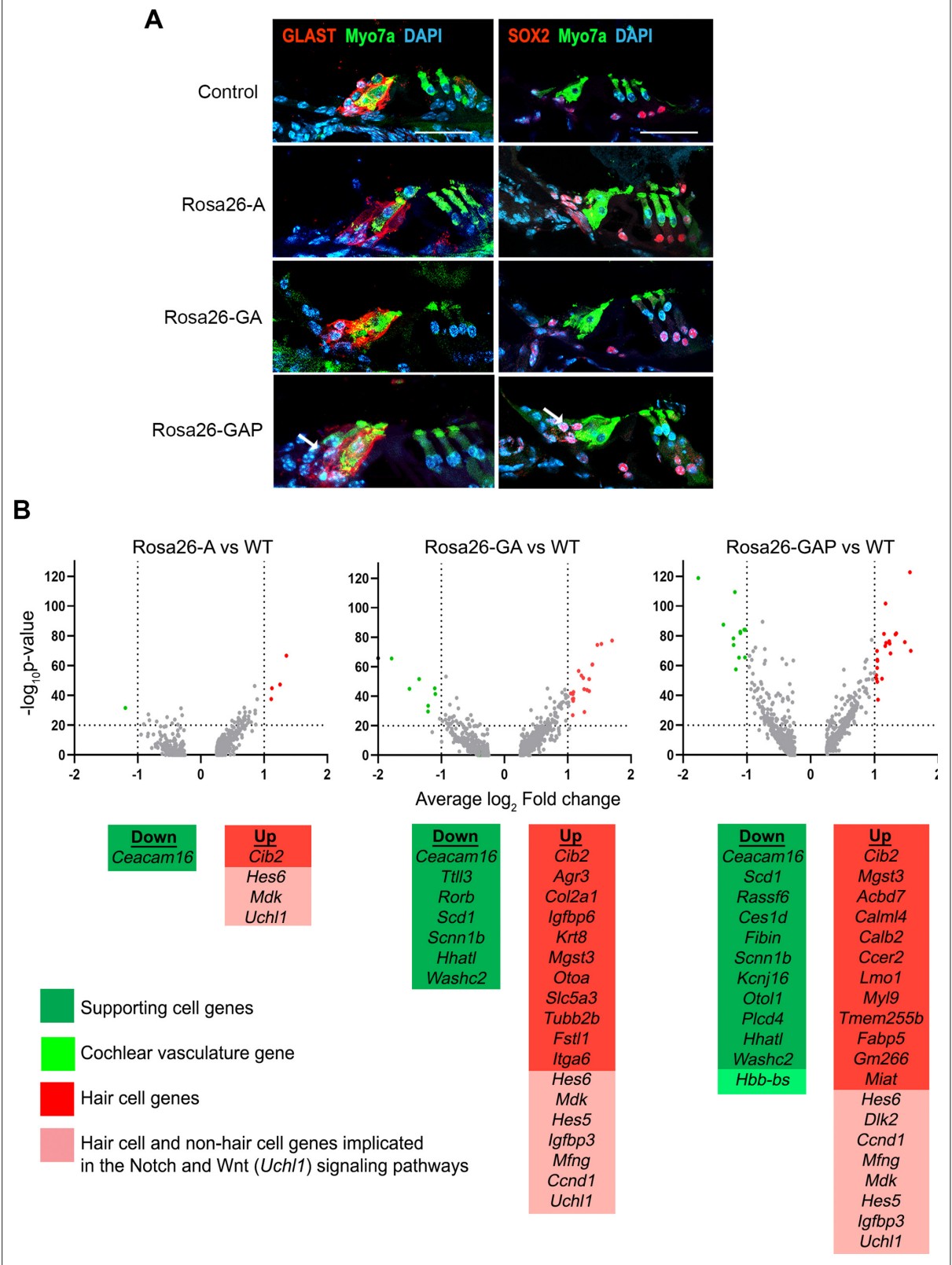

**Figure 8.** Rosa26-GAP reprogramming from day 8 to day 15 induces ectopic GLAST + supporting cells and upregulates some hair cell genes in endogenous supporting cells. (**A**) Control and reprogrammed cochleae immunostained for markers specific to inner phalangeal and border cells (GLAST), a general supporting cell marker (SOX2), and a hair cell marker, Myosin VIIA. A reprogrammed GLAST positive supporting cell in Rosa26-GAP condition is indicated with arrows. Images show a 16 μm section of the organ of Corti (control and reprogrammed). Scale bar: 50 μm. (**B**) Single-cell RNA

*Figure 8 continued on next page*

*Figure 8 continued*

seq analysis of supporting cells under reprogramming conditions (induction at day 8, analysis at day 15). Volcano plots show that several hair cell-specific genes and Notch pathway genes are upregulated by reprogramming factors, while several supporting cell genes are downregulated.

The online version of this article includes the following source data and figure supplement(s) for figure 8:

**Source data 1.** List of genes downregulated in supporting cells in response to transcription factor reprogramming at P15.

**Source data 2.** List of genes upregulated in supporting cells in response to transcription factor reprogramming at P15.

**Figure supplement 1.** Transcription factor induction at P8 does not influence cell proliferation status in the control and reprogrammed cochlea at P15.

also provided a simple mechanistic explanation for why our three reprogramming mice – *Rosa-A, Rosa-GA,* and *Rosa-GAP* – were all equally capable of generating reprogrammed hair cells in neonatal mice. At P1, our multi-omic analysis shows that the *Pou4f3* locus is epigenetically accessible in cells of the GER. We could identify accessible peaks with ATOH1 binding sites in this locus (***Figure 9—figure supplement 1B, C***), suggesting that activation of ATOH1 in the GER at this age could also induce expression of POU4F3. Accordingly, we found that expression of ATOH1 alone induced POU4F3 protein throughout the GER with 3 days after tamoxifen addition (***Figure 9—figure supplement 1A***). However, in 8-day-old mice, the *Pou4f3* locus was significantly less accessible in GER cells (***Figure 9—figure supplement 1B***), suggesting that ATOH1 alone would not be sufficient to activate these factors in GER cells. As expected, we saw no evidence for induction of POU4F3 protein when ATOH1 was activated in Rosa-A mice at P8 and POU4F3 analyzed by immunostaining at P11 (***Figure 9—figure supplement 1A***).

## Discussion

ATOH1 is the first transcription factor to be expressed in differentiating hair cells and is sufficient to generate large numbers of new hair cell-like cells when ectopically expressed in non-sensory regions of the embryonic or neonatal mouse cochlea (***Kelly et al., 2012***; ***Liu et al., 2012***). However, its ability to reprogram these non-sensory cells to a hair cell fate declines in the first postnatal week, prompting attempts to augment its reprogramming activity with combinations of other hair cell transcription factors (***Costa et al., 2015***; ***Lee et al., 2020***; ***Menendez et al., 2020***; ***Walters et al., 2017***; ***Yamashita et al., 2018***). Here, we show that the co-expression of ATOH1 with two other hair cell transcription factors, GFI1 and POU4F3, in *Rosa-GAP* mice can increase the efficiency of hair cell reprogramming in older animals compared to ATOH1 alone or GFI1 +ATOH1. However, the hair cells generated by reprogramming at 8 days of age – even with three hair cell transcription factors – are significantly less mature than those generated by reprogramming at postnatal day 1. By analyzing the epigenetic landscape of the cochlea over the first two postnatal weeks, we suggest that reprogramming with multiple transcription factors is better able to access the hair cell differentiation gene regulatory network, but that additional interventions may be necessary to produce mature and fully functional hair cells.

By targeting different transcription factor combinations to the same locus – *Rosa26* (***Figure 1A***) – we were able to directly compare the reprogramming ability of three hair cell transcription factor combinations without the confounds of variable expression levels caused by different transgene copy numbers or integration sites. Our results show that in newborn mice, activation of the reprogramming cocktails – ATOH1, ATOH1 +GFI1, and ATOH1 +GFI1+POU4 F3 – can produce equally large numbers of new inner hair-cell-like cells in the greater epithelial ridge that receive neuronal input, form ribbon synapses, form immature stereocilia resembling those of endogenous hair cells at this age, and exhibit rudimentary mechanotransduction properties as shown by FM1-43 uptake and expression of components of the mechanotransduction apparatus (***Figures 1 and 2***). Moreover, these new hair-cell-like cells can survive in the transformed greater epithelial ridge for at least two weeks, overriding the process of GER remodeling via apoptosis which occurs during normal cochlear development in mammals (***Figure 1—figure supplement 2***). Consistent with the similar morphological and functional properties of these reprogrammed cells, we found no significant differences in the transcriptomes of the reprogrammed hair-cell-like cells produced by the three transcription factor combinations when induced at P1 and analyzed a week later (***Figure 3***). The simplest explanation for these results is that the *Gfi1* and *Pou4f3* genes are direct transcriptional targets of ATOH1 (***Hertzano et al., 2004***; ***Ikeda et al., 2015***; ***Yu et al., 2021***), and thus activation of either ATOH1 alone (or GFI1+ATOH1) would

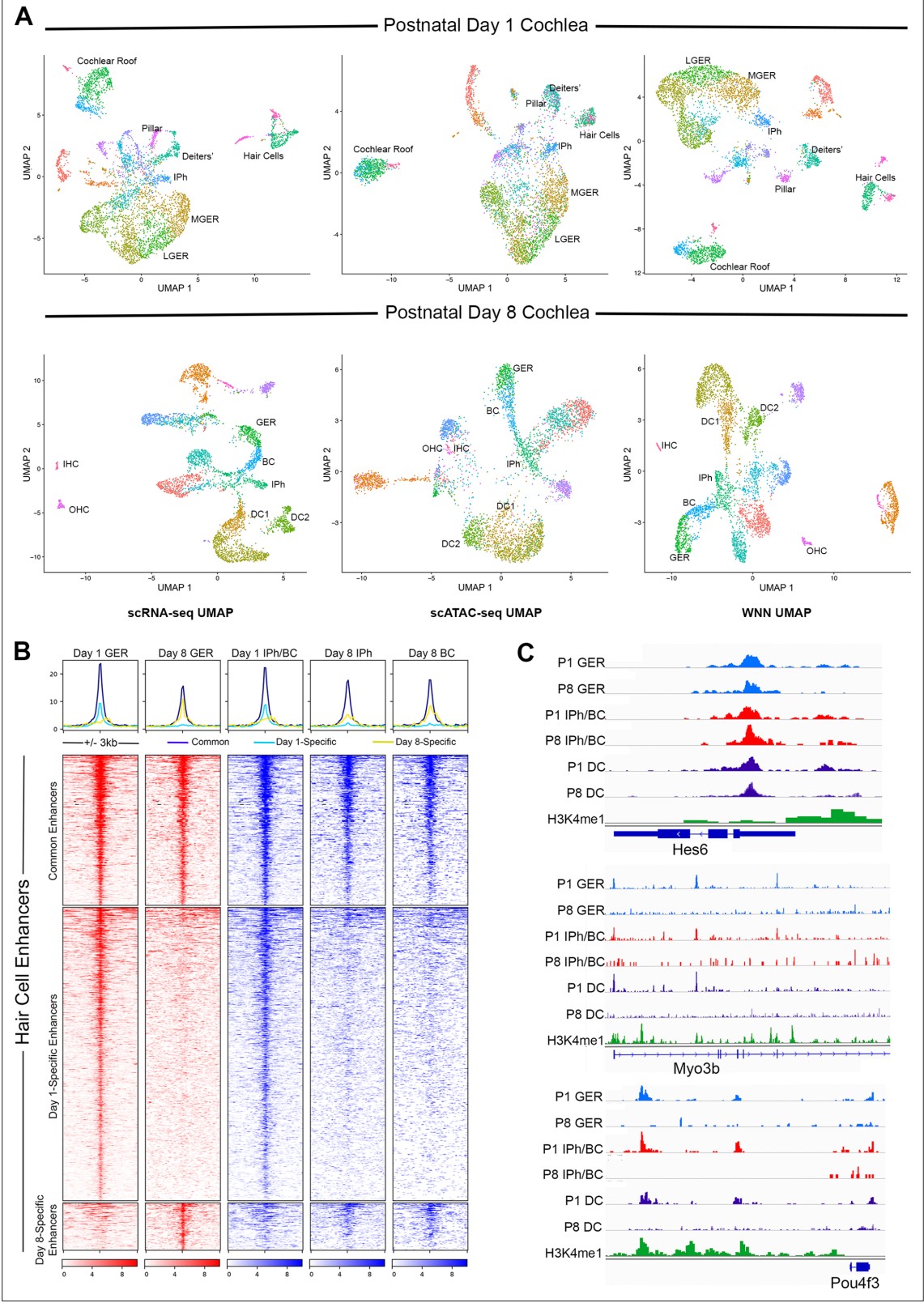

**Figure 9.** Multiomic analysis of 1 and 8-day-old mouse cochlea shows a loss of epigenetic accessibility of hair cell loci in GER and supporting cells. (**A**) Clustering of P1 and P8 cochlear cells on the basis of scRNA-seq, scATAC-seq and weighted-nearest neighbor analyses. Different cochlear cell types can be resolved at both ages. IPh: Inner phalangeal cells; MGER: Medial greater epithelial ridge; LGER: Lateral greater epithelial ridge; IHC: Inner hair cells; OHC: Outer hair cells; GER: Greater Epithelial Ridge; BC: Border cells; DC1 and 2: Deiters' cells. (**B**) Heat map showing ATAC-seq peaks of 1627

*Figure 9 continued on next page*

*Figure 9 continued*

distal regulatory elements identified in hair cell gene loci. ATAC-seq data was extracted from day 1 and day 8 GER cells, and inner phalangeal and border cells. (**C**) Examples of changes in the accessibility of three hair cell loci (*Hes6, Myo3b, Pou4f3*) in GER cells and supporting cells in P1 and P8 mouse cochlea, measured by scATAC-seq. H3K4me1 data for each locus is taken from *Tao et al., 2021*. Reductions in accessibility can be seen in all three loci between P1 and P8.

The online version of this article includes the following figure supplement(s) for figure 9:

**Figure supplement 1.** Endogenous Pou4f3 expression can be induced in 1-day-old, but not 8-day-old cochleae by reprogramming with Atoh1.

result in the activation of all three transcription factors, together with other transcriptional effectors of the hair cell gene regulatory network. In support of this idea, we observe precocious induction of POU4F3 protein in GER cells several days after the expression of ATOH1 in Rosa-A mice. In contrast, activation of ATOH1 alone just one week later in P8 Rosa-A mice is unable to activate POU4F3 expression (*Figure 9—figure supplement 1A*) suggesting that components of the hair cell gene regulatory network – even those immediately downstream of *Atoh1* – become refractory to induction by ATOH1 alone in older animals.

Our data show that co-expression of GFI1 and POU4F3 with ATOH1 in the 8-day-old cochlea is sufficient to generate significant numbers of reprogrammed hair-cell-like cells by P15, but that ATOH1 alone or ATOH1 + GFI1 produce very few new hair cells. It is important to note that the GER is undergoing significant apoptotic remodeling at this time in response to thyroid hormone (*Peeters et al., 2015*), as shown by the loss of Sox9 lineage-labeled cells from the cochlea between P8 and P15 (Figure S4A). This remodeling likely leads to a loss of cells capable of being reprogrammed by *Rosa-GAP* mice, which we suggest leads to an under-estimate of the reprogramming capability of *Rosa-GAP* mice in these experiments. Nonetheless, despite the presence of significant numbers of newly reprogrammed hair cells in *Rosa-GAP* mice, our single-cell RNA-seq analysis of the new hair-cell-like cells at P15 reveals no significant differences in their transcriptome compared to the small number of reprogrammed cells generated by *Rosa-A* and *Rosa-GA* mice (*Figure 6C*). In addition, new hair cells generated in all three conditions between P8 and P15 have only rudimentary mechanotransduction as assayed by FM1-43X uptake (*Figure 5—figure supplement 1*). These data suggest that although the combination of our three transcription factors can significantly increase the *efficiency* of hair cell reprogramming, they are not able to activate additional components of the hair cell gene regulatory network compared to ATOH1 alone or ATOH1 + GFI1 – in other words, the additional reprogramming factors can enhance the quantity of reprogrammed hair cells, but not their 'quality'. However, this conclusion should be qualified by several considerations. First, the number of reprogrammed hair cells appearing in our scRNA-seq analysis of *Rosa-A* and *Rosa-GA* is very small, and second, since our RNA-seq analysis was performed only 1 week after activation of the reprogramming factors at P8, additional maturation may likely occur after longer periods. Consistent with this possibility, we note that Myosin VIIa + cells produced by *Rosa-GAP* mice at P15 do not have organized actin bundles on their apical surface that can be labeled by phalloidin. Significantly, such bundles begin to appear on these GAP-reprogrammed hair cells over the following two weeks, but not in the small number of extra hair cells generated by Rosa-A mice (*Figure 4—figure supplement 1B*). It is also important to note that our current over-expression model causes continued expression of ATOH1 in our reprogrammed cells, whereas *Atoh1* expression is normally downregulated in hair cells shortly after they begin to differentiate. Such persistent expression may militate against full maturation of the new hair cells generated in our studies, and in previous studies that have used constitutive activation of *Atoh1* as a reprogramming strategy (*Chen et al., 2021*; *Kelly et al., 2012*; *Lee et al., 2020*; *Liu et al., 2012*; *Menendez et al., 2020*; *Walters et al., 2017*; *Yamashita et al., 2018*).

The new hair cells generated by reprogramming in the GER region adjacent to the endogenous inner hair cells expressed at least one inner hair cell marker (VGLUT3). However, we saw no evidence for this in our P15 reprogrammed cells, suggesting that developmental signals that may specify inner versus outer hair cells may have disappeared by this age. Reprogramming with additional transcription factors specific for hair cell sub-types – such as Ikzf2 or Insm1 for outer hair cells (*Chessum et al., 2018*; *Wiwatpanit et al., 2018*) or Tbx2 for inner hair cells (*García-Añoveros et al., 2022*) may be necessary to generate specific kinds of hair cells in older animals. Indeed, a recent report suggests that Atoh1 and Ikzf2 can co-operate to produce hair cells with some properties of outer hair cells (*Sun et al., 2021*).

Supporting cells are essential for the function and survival of inner ear hair cells. During development, hair cells and supporting cells derive from a common progenitor, and the correct proportion of these two cell types is regulated by several signaling pathways, most prominently Notch signaling (reviewed by *Basch et al., 2016a*). Interventions that promote conversion of supporting cells into hair cells without replacing the original supporting cells are unlikely to lead to the restoration of a functional organ of Corti. Non-mammalian vertebrates accomplish this by triggering transient proliferation of at least some supporting cells during the regenerative process, and such transient proliferation may be important to promote functional hair cell regeneration in mammals. However, the generation of reprogrammed hair cells can generate new supporting cells in the surrounding tissue through non-cell-autonomous mechanisms, again including Notch signaling (*Stone and Cotanche, 1994*; *Wan et al., 2020*; *Zhang et al., 2018*). In the present study, we have shown that the production of large numbers of reprogrammed hair cell-like cells in the GER causes the induction of GLAST + supporting cells that interleave the new hair cells. Of note, the new hair cells produced in the GER resemble inner ear hair cells, and our scRNA-seq and antibody characterization shows that the identity of ectopic supporting cells resembles border cells and inner phalangeal cells, two supporting cell types that normally surround inner hair cells. Reprogramming in the GER is, therefore, able to generate appropriately patterned and location-specific mosaics of hair cells and supporting cells similar to those that occur in vivo.

The fact that the new supporting cell-like cells retain their identity despite continuing to express hair cell reprogramming factors suggests that signaling pathways present in supporting cells can override the action of the reprogramming factors. Our data also show that some endogenous supporting cells in the organ of Corti – notably pillar cells and Deiters' cells – remain refractory to the effects of reprogramming factors at both P1 and P8. We have confirmed the refractory state of these supporting cell types with two different Cre lines, *Sox9-CreER* and *Lfng-CreER* (*Figure 7—figure supplement 1*). However, a recent study also expressed ATOH1, GFI1, and POU4F3 in different populations of neonatal cochlear supporting cells and reported that some of them are capable of being reprogrammed into hair cells (*Chen et al., 2021*). One of the Cre-expressing lines used to activate the reprogramming factors in the study by Chen and colleagues, *Fgfr3-iCreER^{T2}*, also expresses functional Cre recombinase in up to 30% of outer hair cells at the experimental time points and ages used in the study (*Cox et al., 2012*), so it is likely that many of the labeled outer hair cells were endogenous hair cells present at the start of the experiment. Although the mice generated by *Chen et al., 2021* apparently targeted the three reprogramming factors to the ROSA locus and employed a chicken beta-actin promoter in the same manner as the *Rosa-GAP* mice we report here, it is also possible that the two targeted lines express the reprogramming factors at different levels. Although we observe transcriptional changes in supporting cells in response to our three reprogramming combinations (*Figure 8*), both the hair cells and supporting cells in the organ of Corti remain healthy and viable in all conditions examined (*Figures 1–8*, S1-S9). In contrast, hair cell loss was observed when ATOH1, GFI1, and POU4F3 were activated by either *Fgfr3-iCreER^{T2}* or *Lgr5-CreER* mice (*Chen et al., 2021*), suggesting that the levels of reprogramming factors used may be important for cochlear cell viability. Hair cell survival appears to be particularly sensitive to ATOH1 levels; hair cell loss and hearing deficits are observed in mice with only one functional ATOH1 allele, or with two hypomorphic alleles of ATOH1 (*Xie et al., 2017*), and so it is likely that future regenerative strategies using ATOH1 will need to calibrate the level and duration of this important transcription factor with great precision.

Several lines of evidence suggest that Notch signaling may be responsible for overriding the reprogramming ability of our transcription factor combinations. First, most organ of Corti supporting cells in neonatal mice rapidly and readily trans-differentiate into hair cells when Notch signaling is blocked (*Jiang et al., 2014*; *Korrapati et al., 2013*; *Mizutari et al., 2013*). Second, blocking Notch signaling with gamma-secretase inhibitors can cause ectopic supporting cells in the GER created by ATOH1 reprogramming to transdifferentiate to hair cells (*Kelly et al., 2012*). Third, our scRNA-seq data suggests that elements of the Notch signaling pathway are reconstituted in the ectopic supporting cells generated in our mice (*Figure 7*), and in endogenous supporting cells that receive reprogramming factors from P8-P15 (*Figure 8*). This latter result is particularly notable, as supporting cells normally transcriptionally and epigenetically down-regulate the Notch pathway in the first postnatal week and become refractory to Notch inhibition (*Maass et al., 2015*; *Maass et al., 2016*; *Tao et al., 2021*). Together, our data suggest that transcription factor reprogramming may reconstitute some

of the transcriptional and epigenetic regulation that normally exists between developing hair cells and supporting cells, and it will be of interest in the future to understand how fully these regulatory circuits can be established by different combinations of transcription factors. In addition to the effects of Notch signaling, our multi-omic analysis of cochlear tissue at P1 and P8 shows clearly that the chromatin of hair cell loci becomes less accessible in supporting cells and GER cells during the first postnatal week (*Figure 9*). This decrease in accessibility affects hair cell loci in general, but significantly is also seen in key hair cell transcription factors downstream of Atoh1, including *Gfi1* and *Pou4f3* (*Figure 9—figure supplement 1A*). By using a single-cell multi-omic analysis, we were able to demonstrate these epigenetic changes occurring in all supporting cell and GER cell populations during the first postnatal week. We suggest that this decrease in accessibility is a second element contributing to the need for multiple transcription factors to reprogram more mature cochlear tissue.

In conclusion, our work shows that while overexpression of multiple hair cell transcription factors in the cochlea clearly promotes more efficient reprogramming in older animals, significant challenges to producing viable, functional hair cells still remain. Future work will be necessary to determine whether more functional hair cells can be generated with extra hair cell transcription factors, by epigenetic modulation of hair cell loci in supporting cells, or by actively targeting the down-regulation of supporting cell genes during reprogramming. Finally, we emphasize that our present work focuses exclusively on the intact organ of Corti. We currently do not know what effect the acute and long-term pathological consequences of hair cell loss in the cochlea will have on the efficiency and fidelity of hair cell reprogramming, and addressing this question will be critical to promoting functional hair cell regeneration in the mammalian cochlea.

# Materials and methods

## Targeting of the ROSA locus

The three conditional lines for transcription factor overexpression (Rosa-A, GA, GAP) were constructed by modifying the *Ai3* targeting construct (Addgene #22797; *Madisen et al., 2010*). The EGFP insert in *Ai3* was removed by FseI digestion and replaced with coding regions for the following: *Rosa-A: Atoh1* fused to EGFP (*Rose et al., 2009*); *Rosa-GA: Gfi1* and *Atoh1-EGFP* separated by a GSG-P2A sequence; *Rosa-GAP: Gfi1, Atoh1-EGFP,* and *Pou4f3* separated by GSG-P2A sequences. The targeting constructs were digested with PacI and AscI to separate the construct from the homology arms and cloned into a p15a-based targeting vector containing homology arms for the ROSA26 locus (5′: 1057 bp; 3′: 1231 bp). Linearized targeting constructs (2 µg) were electroporated into AB2.2 ES cells with 20 µg of a pX330 plasmid (Addgene # 42230) expressing Cas9 and a sgRNA sequence to target the ROSA26 locus just inside the 5′ homology arm: ACTGGAGTTGCAGATCACGA *GGG* (PAM sequence is shown in italics). Forty-eight neomycin-resistant clones were picked, verified for correct targeting of the ROSA26 locus, expanded, and injected into 129 blastocysts to create chimeras, which were then bred to C57Bl6 mice to establish germline founders.

## Experimental animals

All mouse experiments were performed at Baylor College of Medicine and approved by the Institutional Animal Care and Use Committee (IACUC). In addition to the *Rosa26*-targeted mice described above, we also used several lines available from the Jackson Laboratory: *Sox9-CreER^T2* mice (*Tg(Sox9-CreERT2)1Msan/J*; stock# 018829; RRID:IMSR_JAX:018829), *Lfng-CreER* mice (*Tg(Lfng-cre/ERT2)1Mmsa/J*; stock# 035554; RRID:IMSR_JAX:035554), *Ai3* EGFP Cre reporter mice (*Cg-Gt(ROSA)26Sor^tm3(CAG-EYFP)Hze/J*; stock #007903; RRID:IMSR_JAX:007903) and *Ai9* tdTomato Cre reporter mice (*Cg-Gt(ROSA)26Sor^tm9(CAG-tdTomato)Hze/J*; stock# 007909; RRID:IMSR_JAX:007909). *Ai3* and *Ai9* mice are described in *Madisen et al., 2010*. For single-cell RNA sequencing work, we incorporated the *Ai9* reporter allele into our three types of crosses to yield mice of the genotypes *Sox9-CreER^T2:Rosa26^Atoh1/Gfi1-Atoh1/Gfi1-Atoh1-Pou4f3; Rosa26^tdtomato*. Experimental animal genotypes for all other work were *Sox9-CreER^T2; Rosa26^Atoh1/Gfi1-Atoh1/Gfi1-Atoh1-Pou4f3*.

## Mouse genotyping

The following primer pairs were used for genotyping:

*Sox9-CreER^T2* mice and *Lfng-CreER* mice: Forward primer – (GCC TGC ATT ACC GGT CGA TGC AAC GA), reverse primer – (GTG GCA GAT GGC GCG GCA ACA CCA TT) yielding a band of 700 bp.

## Ai3 EGFP and Ai9 tdTomato Cre reporter mice

*Wild type forward primer* (AAG GGA GCT GCA GTG GAG TA), wild type reverse primer – (CCG AAA ATC TGT GGG AAG TC), mutant forward primer – (ACA TGG TCC TGC TGG AGT TC), mutant reverse primer (GGC ATT AAA GCA GCG TAT CC) yielding a wild type band of 297 bp and a mutant band of 212 bp (https://www.jax.org/Protocol?stockNumber=007903&protocolID=28710). EGFP could also be detected with forward primer – (CGA AGG CTA CGT CCA GGA GCG CAC), reverse primer – (GCA CGG GGC CGT CGC CGA TGG GGG TGT) yielding a band of 300 bp.

*ROSA* modified reprogramming mouse alleles: The wild type *Rosa26* allele was detected using the wild type primers for *Ai3* and *Ai9* listed above, yielding a band of 297 bp. The *Rosa-A* allele was specifically detected with forward primer – (AAA TGA CCA CCA TCA CCT TCG CAC C) and reverse primer – (ACG CTG AAC TTG TGG CCG TTT ACG TC), yielding a band of 483 bp. The *Rosa-GA* allele was specifically detected with forward primer – (ACA TCT GCT CAT TCA CTC GGA CAC C) and reverse primer – (TTT ACC TCA GCC CAC TCT CT GCA TG), yielding a band of 384 bp. The *Rosa-GAP* allele was specifically detected with forward primer – (CTA TTT CGC CAT CCA GCC ACG TCC TTC) and reverse primer – (GAC AAC GGG CCA CAA CTC CTC ATA AAG), yielding a band of 375 bp.

## Tamoxifen treatment

Tamoxifen (Sigma Aldrich) was dissolved in peanut oil at a concentration of 20 mg/ml. This solution was volume optimized and injected subcutaneously at a dosage of 0.2 mg/g body weight into P1 and P8 animals. Experimental and control littermates were genotyped and segregated after harvest.

## Western blotting

Cells were lysed in lysis buffer (50 mM Tris-HCl, pH 7.5, 150 mM NaCl, 0.5% Triton X-100, 5% glycerol, 1% SDS, 1 x protease inhibitor cocktail). Protein concentrations were determined using a BCA assay kit (Bio-Rad). Ten µg of protein lysate was boiled with 6 X SDS sample buffer (0.5 M Tris-HCl pH 6.8, 28% glycerol, 9% SDS, 5% 2-mercaptoethanol, 0.01% bromophenol blue) and electrophoresed on a 4–15% Criterion Tris-HCl gel (Bio-Rad) and transferred onto a PVDF membrane. Membranes were blocked for 1 hr at room temperature or overnight at 4 °C using blocking buffer (5% milk in TBST). Following blocking, membranes were incubated with appropriate dilutions of primary antibody (GFP 1:500 (Santa Cruz), ATOH1 1:1000 (Proteintech), GFI1 1:1000 (Abcam), POU4F3 1:500 (Santa Cruz)) in blocking buffer for overnight 4 °C on a rocker. Next, membranes were washed three times in TBST, 5 min each at room temperature. After this, membranes were incubated with the recommended dilution of conjugated secondary antibody in blocking buffer at room temperature for 1 hr. Membranes were washed 3 times in TBST, 5 min each. The signals were developed using Immobilon Western Chemiluminescent HRP Substrate (Millipore) and detected using ImageQuant LAS 4000 (GE Healthcare) according to the manufacturer's instructions.

## Fixation, dissection, and cryosectioning

Temporal bones from P8 and P15 mice were harvested and fixed in 4% paraformaldehyde for 2 hr at room temperature on a shaker. Fixed temporal bones were stored in 1 X PBS at 4 °C and micro-dissected with fine forceps to peel out the cochlear epithelium. In some cases, P15 temporal bones were decalcified in 0.3–0.5 M EDTA (pH 8.0) for 3–4 hr at room temperature. For cryosectioning, samples were immersed in a 15% sucrose (Fisher Bioreagents #141913) solution at 4 °C overnight. The temporal bones were then incubated for two hours in a sucrose-gelatin solution (7.5% gelatin (Sigma SLBX 2973) /15% sucrose and 0.0025 mg of sodium azide in 1 X PBS, dissolved at 65 °C and stored at 37 °C), followed by embedding and sectioning to give 12–14 µm serial sections on a Leica CM 1850 cryostat.

## Immunohistochemistry

Whole cochlear epithelia were permeabilized with 0.5% Triton-X in 1 X PBS at room temperature for 20 min. Sections were subject to gelatin removal by incubating in 1 X PBS at 37 °C for 10 min followed by washing. Note - The mouse Myosin VIIA and Rat SOX2 antibodies require a specific antigen

retrieval step at this point. The slides/tissues were incubated in the antigen retrieval solution (10 mM sodium citrate solution made by dissolving sodium citrate salt or citric acid powder in distilled water. The pH of this solution is adjusted to 6.0 using conc.HCl or NaOH, respectively. 0.05% Tween 20 is added and dissolved to get a clear solution) for 15 min at 80 °C. The samples were cooled to room temperature without replacing the solution and washed three times. Sections were permeabilized with 0.3% Triton-X in 1 X PBS at room temperature for 5 min. Post permeabilization, tissues (whole cochlear epithelium and sections) were washed three times with 1 X PBS for 10 min each. Tissues were blocked with 10% goat serum for 1 hr at room temperature. Primary antibody combinations were diluted in 5% goat serum with 0.2% Triton-X and incubated overnight at 4 °C. Note- For the rabbit anti-PROX1, blocking and antibody dilutions were in 10% and 1% donkey serum along with Triton X-100 respectively. For the anti-CTBP2 staining, primary antibody incubation was at 37 °C overnight in a humidified chamber. After three washes with 1 X PBS, tissues were incubated with fluorescently labeled secondary antibodies diluted in 5% goat serum with 0.2% Triton-X or 1 X PBST and incubated for 2 hr at room temperature. Tissues were counterstained with DAPI (1:5000), washed, and dried. Tissues were mounted using the Fluormount (Southern Biotech) mounting medium, sealed with glass coverslips, and dried before imaging.

## Antibodies

| Antigen | Host | Source | RRID | Dilution |
|---|---|---|---|---|
| Myosin VIIa | Rabbit polyclonal | Proteus Biosciences 25–6790 | AB_10015251 | 1:300 |
| Myosin VIIa | Mouse polyclonal | DSHB 138–1 | AB_2282417 | 1:200 |
| TUJ1 | Mouse polyclonal | BioLegend 801213 | AB_2728521 | 1:1000 |
| VGLUT3 | Rabbit polyclonal | Synaptic systems 135203 | AB_887886 | 1:300 |
| PRESTIN | Rabbit polyclonal | Gift from Dr.Jing Zheng | AB_2315199 | 1:1000 |
| EpCAM/CD326 | Rat polyclonal | eBioscience 17-5791-80 | AB_2734965 | 1:300 |
| POU4F3 | Rabbit polyclonal | Proteintech 21509–1-AP | AB_2878872 | 1:200 |
| GLAST | Rabbit polyclonal | Abcam ab416 | AB_304334 | 1:300 |
| SOX2 | Rat monoclonal | Biocompare 14-9811-80 | AB_11219070 | 1:250 |
| SOX9 | Rabbit polyclonal | Millipore Sigma AB5535 | AB_2239761 | 1:200 |
| PROX1 | Rabbit polyclonal | Millipore Sigma AB5475 | AB_177485 | 1:300 |
| CTBP2 | Mouse IgG1 | BD Biosciences 612044 | AB_399431 | 1:300 |
| RFP | Chicken polyclonal | Millipore Sigma AB3528 | AB_91496 | 1:300 |
| GFP | Chicken polyclonal | Abcam ab13970 | AB_300798 | 1:500 |
| AF-488 goat anti rabbit IgG (H+L) secondary | | Thermo Fisher Scientific (Invitrogen) A-11008 | AB_143165 | 1:500 |
| AF-488 goat anti-mouse IgG (H+L) secondary | | Thermo Fisher Scientific (Invitrogen) A-11001 | AB_2534069 | 1:500 |

*Continued on next page*

*Continued*

| Antigen | Host | Source | RRID | Dilution |
|---------|------|--------|------|----------|
| AF-594 goat anti rabbit IgG (H+L) secondary | | Thermo Fisher Scientific (Invitrogen) A-11012 | AB_2534079 | 1:800 |
| AF-594 goat anti-mouse IgG (H+L) secondary | | Thermo Fisher Scientific (Invitrogen) A-11005 | AB_2534073 | 1:500 |
| AF-647 goat anti-mouse IgG1 secondary | | Thermo Fisher Scientific (Invitrogen) A-21240 | AB_141658 | 1:500 |
| AF-488 goat anti rat IgG (H+L) secondary | | Thermo Fisher Scientific (Invitrogen) A-11006 | AB_2534074 | 1:500 |
| AF-594 goat anti chicken IgY(H+L) secondary | | Thermo Fisher Scientific (Invitrogen) A-11042 | AB_2534099 | 1:500 |
| Phalloidin 488 | | Thermo Fisher Scientific (Invitrogen) A-12379 | AB_2631056 | 1:1000 |

## Microscopy and image processing

Immunostained samples were viewed on an LSM 780 confocal microscope in the Baylor Optical Imaging & Vital Microscopy Core at 20 X and with emulsion oil applied to a 40 X objective lens. Exposure levels were maintained between slides that were part of the same experimental batch. Maximum intensity projections obtained after processing z-stacks were processed using Adobe Photoshop CS6. Processing steps include normalization of intensity levels and derivation of 200 μm lengths of the tissue measured using the scale bar option on the Zen Blue 3.1 software.

## Cell number quantification

All the images that were used for cell counting were analyzed on Zen Blue 3.1. Counting was done by using the event and marker options that sum the number of objects clicked upon. Inner hair cells, outer hair cells, and GER hair cells from 3 images were counted per genotype – WT, *Rosa-A, Rosa-GA*, and *Rosa-GAP*. Bar graphs were plotted employing the Graphpad Prism 5.0 software after performing an unpaired t-test comparing the control and each induced condition, to determine significance. For all experiments, three biological replicates (i.e. parallel measurements of biologically distinct samples) were used.

## Scanning electron microscopy

Fixing solution (all reagents from Electron Microscopy Solutions) was prepared by mixing 8% glutaraldehyde (2% final concentration), 0.6 M Cacodylate buffer (pH 7.2–7.4; 0.15 M final concentration) and distilled water. The temporal bones from P8 and P15 experimental animals were removed, and the apex region of each sample was punctured. The temporal bones were incubated in the SEM fixing solution for 2 hr at room temperature. Post incubation, the temporal bones were rinsed and stored in 0.6 M cacodylate buffer at 4 °C. Samples were micro-dissection to expose the organ of Corti and processed with the OTOTO method for scanning electron microscopy. Tissues were then dehydrated in graded ethanol solutions, critical point dried, and mounted on a stub using silver paste. Images were taken with a TESCAN Rise scanning electron microscope.

## FM 1-43 dye uptake assay

FM 1–43 dye solution was prepared by dissolving a 10 μg/μl stock of FM 1–43 (Thermo Fisher, Cat no. T35356) dye in EBSS/HBSS to a final concentration of 2 μg/ml in 0.5 ml. One cochlea at a time was dissected from live P8 experimental mice and placed on a glass slide. The tissue was incubated in 20 μl of the dye solution for 10–12 s and immediately washed with 1 X PBS. The orientation of the tissue was checked followed by the addition of mounting medium. The mounted tissue was sealed

with a glass coverslip and imaged under the 488 (green) channel of a Zeiss fluorescence microscope. The light intensity and brightness were normalized for images captured from different samples using the endogenous hair cells as a reference. For labeling of P15 animals, temporal bones were dissected from the skull and the semicircular canals removed. Using a 30 gauge needle, a fenestra was created at the helicotrema and FM1-43 FX was injected into the cochlear duct and incubated for ~30 s. To remove the FM 1–43 FX, 4% paraformaldehyde (16% stock, 18814–10, Polysciences) was injected into the cochlear duct and cochlea were fixed overnight at 4 °C. Fixed cochlea were decalcified with 0.25 M EDTA for 6 hr at room temperature, followed by dissection of the sensory epithelium. Tissue was mounted in ProLong Gold (Invitrogen) and imaged with a Zeiss LSM 880 confocal microscope.

### Cell proliferation assay and EdU staining

An EdU injection solution of 5 mg/ml concentration was prepared by dissolving EdU powder (Thermo Fisher Scientific, #A10044) in 1 X PBS pH 7.4 (Thermo Fisher Scientific, #10010023). Pups were weighed and injected with this solution at a dose of 50 mg/kg body weight, subcutaneously. Injections were given twice every alternate day (9 am and 5 pm). Mice at P8 and P15 were collected, fixed, and cryosectioned following the procedures described above for IHC. EdU visualization was done using the Click-iT EdU Cell Proliferation Kit for Imaging, Alexa Fluor 488 dye (Thermo Fisher Scientific, #C10337) following the manufacturer's instruction accompanying the kit. Co-immunostaining with primary antibody for Myosin VIIA was performed as described previously in this section.

### Single-cell dissociation of cochlear cells and FACS purification

Whole ears from P8 mice were harvested and transferred to a dish of ice-cold CMF PBS. The organ of Corti was dissected from controls and experimental samples into separate microcentrifuge tubes with 0.3 ml ice-cold CMF PBS placed on ice. For P15 animals, whole temporal bones were cleaned and punctured at the apex, then placed into tubes with 0.3 ml ice-cold CMF PBS. In both cases, tissue was washed twice with 0.3 ml ice-cold CMF PBS and incubated in 0.3 ml papain solution (20 U/ml, 1 mM L-Cysteine and 0.5 mM EDTA; Worthington Biochemical Corporation) at 37 °C for 40 min. The papain solution was removed carefully, and the tissue was washed twice with 0.3 ml ice-cold CMF PBS containing 2–5% FBS. The tissue was triturated for 3–4 min by placing tubes on ice with minimal frothing, then filtered (40 micron then 35 micron filter caps – Pluriselect) into a 5 ml round bottom polystyrene tube to remove clumps and bone fragments before sorting. Dissociated cells were sorted on an AriaII FACS sorter at the BCM Flow Cytometry core. The presort conditions specified for the nozzle were 20 psi pressure and 100 μm size, respectively. TdTomato positive sorted cells were collected in DMEM with 5% FBS solution for cDNA library preparation and single-cell RNA sequencing.

### cDNA library preparation

Purified cells were counted to estimate concentration and loaded onto a 10 X genomics Chromium controller to prepare single-cell 3' RNA seq libraries using the Chromium single cell 3' reagent kit v3 (10 x Genomics). In brief, single cells were partitioned into GEMS (Gel Beads-In-Emulsions) that contain a gel bead with primers that include a Illumina Truseq Read 1 primer, a 16 nucleotide (nt) 10 x barcode, a 12 nt unique molecular identifier and a 30 nt poly(dT) sequence and all the necessary components to perform reverse transcription. Almost simultaneously, the gel bead is dissolved and the partitioned cell is lysed releasing all the cellular RNA. Incubation of these components inside the GEM results in synthesis of full-length barcoded cDNA from the mRNA. Subsequently, the GEMS are lysed and cDNA from all the single-cells are pooled. Following cleanup using Dynabeads MyOne Silane Beads (Thermo Fisher, 370020), cDNA was amplified by PCR and fragmented to optimal size before end-repair, A-tailing, and adaptor ligation to prepare the paired end illumina libraries followed by a final PCR to amplify the library.

### Single-cell RNA sequencing and analysis

Sample QC was conducted using the NanoDrop spectrophotometer and Agilent Bioanalyzer 2100 (High Sensitivity DNA Chip, p/n 5067–4626). To quantitate the adapter-ligated library and confirm successful P5 and P7 adapter incorporations, the Applied Biosystems ViiA7 Real-Time PCR System and a KAPA Illumina/Universal Library Quantification Kit (p/n KK4824) were used. All samples were pooled at equimolar amounts and re-quantitated by qPCR and re-assessed on the Bioanalyzer.

Using the concentration from the ViiA7 TM qPCR machine above, 300pM of the pooled library was loaded onto a NovaSeq S1 flowcell (Illumina p/n 20012865), using the Standard Workflow loading conditions designated by the manufacturer and amplified by exclusion amplification (ExAMP) for patterned flowcells using the Illumina NovaSeq 6000 sequencing instrument. PhiX Control v3 adapter-ligated library (Illumina p/n FC110-3001) was spiked-in at 1% by weight to ensure balanced diversity and to monitor clustering and sequencing performance. A paired-end run, using 28 cycles for Read 1, 8 cycles for Index 1 Read and 91 cycles for Read 2 was set to achieve a minimum of approximately 300 M reads per sample. FastQ file generation and QC assessment were achieved using the 10 X Cell Ranger software for 10 X Chromium Platforms. Sequencing data has been uploaded to the GEO database (https://www.ncbi.nlm.nih.gov/geo), accession number GSE182202.

The unique molecular identifier (UMI) count matrices were generated by aligning the raw reads to the mm10 (GRCm38) genome along with the annotation gtf file (GRCm38realease-93) (from Ensembl) using the count function of the 10 x genomics Cell Ranger pipeline. Alignment, filtering, barcode, and UMI counting were also performed with Cell Ranger. The R package Seurat (v3.2) was used to process the count matrices. First, the count matrices were transformed into Seurat objects, and cells expressing less than 200 genes and genes expressed in less than 5 cells were filtered and excluded from the analysis. Another round of filtration was performed based on the distribution of the nUMI, nFeature, and percentage of mitochondrial genes expressed per cell in each dataset. Cells expressing less than 750 genes and more than 5000 genes as well as cells with greater than 30% mitochondrial genes were excluded from further analysis. The number of cells analyzed per genotype at each time-point is given in the table below.

| Age | Wildtype | Rosa26-A | Rosa26-GA | Rosa26-GAP |
|-----|----------|----------|-----------|------------|
| P8 | 5079 | 8179 | 3097 | 8957 |
| P15 | 4614 | 4015 | 840 | 3758 |

Each dataset was normalized using the logNormalize function, which divided the gene counts for each cell by its total counts; followed by the identification of the top 2000 variable genes using the FindVariableFeatures function.

To identify clusters that differed between the wildtype and the transcription factor-induced data-sets, an integrated analysis of the cells of all the genotypes was performed. First, we identified the integration anchors for the four datasets and used them to integrate the datasets using the Integra-teData function. Next, the integrated dataset was scaled by multiplying the normalized values by a factor of 10,000, followed by dimensionality reduction by principal component analysis (PCA). Top 40 principal components were chosen as significant based on a Jackstraw plot and used to construct a shared nearest neighbor (SNN) graph using the FindNeighbors function. Cells were then clustered at various resolutions ranging from 0.2 to 1.0 using the Leiden algorithm in the FindClusters function of Seurat. The conserved gene markers for each cluster across the different genotypes were identified with the FindConservedMarkers function and unique gene markers for a given cluster were identified using the FindMarkers or for all clusters with FindAllMarkers function. The FindClusters and Find-Markers functions were iterated at the different resolutions until clusters with biological relevance and expected cell types were observed at a resolution of 0.8.

Each cell-type-specific cluster was identified by ranking differentially expressed genes based on the p-value of expression, average log fold change of expression, and the difference in pct1 vs pct2. A search for cell type-specific expression of the top-ranked genes yielded results unique to cell types and was thus labeled. Gene ontology analysis involved the selection of all significantly expressed genes in the hair cell clusters of the P8 and P15 datasets with a cut-off p-value of less than $1.00E^{-03}$. The analysis was done on DAVID (https://david.ncifcrf.gov/tools.jsp) with this gene list as input. Resul-tant GO terms and associated p-values (cut off $1.00E^{-02}$) are represented in the figure. Volcano plots for all differentially expressed genes were plotted using GraphPad Prism 5.0. All cutoffs assigned are marked with dotted lines on the plots.

## Multiomic (combined scRNA- and ATAC-seq) sample processing

For Single Cell Multiome ATAC +Gene Expression (10 x Genomics) experiments, wildtype mice in a mixed background of CD1 and FVB/NJ, and C57BL/6 were used. Cochlear tissue was dissected from P1 and P8 mice and enzymatically dissociated with a cocktail of 400 µl of 0.25% Trypsin, 50 µl of 1 mg/ml Dispase, and 50 µl of 1 mg/ml collagenase for 20 min at 37 °C. After incubation, the digested tissue was titurated 100 times using a small-bore 200 µl pipette and centrifuged at 500 x g for 5 min at 4 °C. After centrifugation, the sample was processed according to the 10 x Genomics protocol 'Nuclei Isolation from Complex Tissues for Single Cell Multiome ATAC +Gene Expression Sequencing' (CG000375, Rev B). Briefly, the supernatant was decanted, and the cell pellet was resuspended in 1 ml of NP-40 Lysis Buffer and incubated for 5 min on ice. Nuclei were filtered through a 40 micron cell strainer, and centrifuged at 500 x g for 5 minutes at 4 °C. The supernatant was decanted, and nuclei were resuspended in 1 ml of PBS +2% BSA and incubated for 5 min at 4 C. Nuclei were centrifuged at 500 x g for 5 min at 4 °C, resuspended in 100 µl of 0.1 x Lysis Buffer, and mixed with a pipet. Nuclei were incubated for 2 min on ice, followed by the addition of 1 ml Wash Buffer, mixed and centrifuged at 500 x g for 5 min at 4 °C, and resuspended in the appropriate volume of Diluted Nuclei Buffer for input into the Single Cell Multiome ATAC +Gene Expression protocol (10 x Genomics). For the P1 cochlea, 6065 nuclei were loaded. For the P8 cochlea, 9645 nuclei were loaded.

## Multiomic data processing

Raw sequencing data from both RNA and ATAC libraries in fastq format were used as input into cellranger-arc count (10 x Genomics, v2.0.0) for simultaneous alignment against the mouse mm10 genome. The cellranger-arc output files 'filtered_feature_bc_matrix.h5' and 'atac_fragments.tsv.gz' were used as input into Seurat v4.1.0 for standard quality control pre-processing, resulting in 4,882 nuclei post-filtering for the P1 dataset, and 7,049 nuclei post-filtering for the P8 dataset. ATAC peaks were called using macs2 v2.1.2. Multiome datasets were clustered based on RNA, ATAC, and weighted-nearest neighbor (*Hao et al., 2021*) to generate UMAPs. This resulted in 20 clusters for the P1 dataset, and 19 clusters for the P8 dataset. Clusters were assigned cell types based on known cell markers. Cell barcodes from clusters of interest (P1 GER, P8 GER, P1 IPh/BC, P8 IPh, and P8 BC) were used to extract ATAC reads belonging to each respective cell type and generate a pseudobulk ATAC dataset. Peaks were called on the pseudobulk ATAC datasets and common peaks were used as input into DESeq2 v1.34.0 to scale the signal between P1 and P8 datasets. Representative signal tracks were visualized in IGV v2.4.14. ATAC peaks were filtered by hair cell enhancers previously identified (*Tao et al., 2021*), and intersected to find common, P1-specific, and P8-specific regions. Heatmaps were generated using deepTools v3.2.0 computeMatrix and plotHeatmap.

## Materials availability statement

The three *Rosa-26* targeted mouse lines (*Rosa-A, Rosa-GA* and *Rosa-GAP*) are available from the corresponding author upon request.

# Acknowledgements

We acknowledge members of the Groves lab for their help and advice, particularly Alyssa Crowder for technical assistance. We thank Gabriela Perez from the Jankowsky lab for help with CTBP2 imaging. We thank Jing Zheng from Northwestern University for providing the Prestin antibody. We acknowledge Teng-Wei Huang for advice with immunostaining protocols. We dedicate this work to the memory of our friend and colleague, Neil Segil, who conceived and oversaw this project with Andy Groves. but sadly passed away before publication. The project was supported by the following grants: RO1 DC014832 (AKG; subcontract to YR), R21 OD025327 (RSR), DC015829 (NS), and a Hearing Restoration Project Consortium award from the Hearing Health Foundation (AKG and NS). The project was also supported by the following cores at Baylor College of Medicine: the Optical Imaging & Vital Microscopy Core with assistance from Jason Kirk; the Cytometry and Cell Sorting Core with funding from the CPRIT Core Facility Support Award (CPRIT-RP180672), and the NIH (P30 CA125123 and S10 RR024574) and assistance from Joel M Sederstrom, Bethany Tiner and Amanda White; the Single Cell Genomics Core, with instrument grants from the NIH (S10OD018033, S10OD023469, S10OD025240) and P30EY002520 and assistance from Rui Chen

and Yumei Li; the Genomic and RNA Profiling Core with assistance from Daniel Kraushaar; the Genetically Engineered Rodent Models core with assistance from Lan Liao and Isabel Lorenzo. The scanning electron microscopy (SEM) work was supported by the R Jamison and Betty Williams Professorship. The SEM unit receives financial support from the University of Michigan College of Engineering and NSF grant #DMR-1625671, and technical support from the Michigan Center for Materials Characterization.

## Additional information

### Competing interests

Hsin-I Jen: Hsin-I Jen is affiliated with Ultragenyx. The author has no financial interests to declare. Jenny J Sun: Jenny J. Sun is affiliated with Moderna. The author has no financial interests to declare. The other authors declare that no competing interests exist.

### Funding

| Funder | Grant reference number | Author |
| --- | --- | --- |
| National Institute on Deafness and Other Communication Disorders | RO1DC014832 | Andrew K Groves Yehoash Raphael |
| National Institute on Deafness and Other Communication Disorders | RO1DC015829 | Neil Segil |
| Hearing Health Foundation | HRP Hearing Restoration Project Consortium Grant | Neil Segil |
| Hearing Health Foundation | Hearing Restoration Project Consortium Grant | Andrew K Groves |
| Office of the Director, National Institutes of Health | R21OD025327 | Russell S Ray |

The funders had no role in study design, data collection and interpretation, or the decision to submit the work for publication.

### Author contributions

Amrita A Iyer, Formal analysis, Investigation, Methodology, Writing – original draft, Writing – review and editing; Ishwar Hosamani, Data curation, Formal analysis, Investigation, Writing – review and editing; John D Nguyen, Formal analysis, Investigation, Methodology, Writing – review and editing; Tiantian Cai, Formal analysis, Validation, Investigation; Sunita Singh, Validation, Investigation, Writing – review and editing; Melissa M McGovern, Investigation, Visualization, Methodology; Lisa Beyer, Investigation, Visualization; Hongyuan Zhang, Hsin-I Jen, Rizwan Yousaf, Onur Birol, Validation, Investigation; Jenny J Sun, Methodology; Russell S Ray, Supervision, Methodology, Writing – review and editing; Yehoash Raphael, Supervision, Investigation, Writing – review and editing; Neil Segil, Conceptualization, Supervision, Funding acquisition, Project administration, Writing – review and editing; Andrew K Groves, Conceptualization, Supervision, Funding acquisition, Investigation, Writing – original draft, Project administration, Writing – review and editing

### Author ORCIDs

Russell S Ray http://orcid.org/0000-0001-9610-2703
Neil Segil http://orcid.org/0000-0002-0441-2067
Andrew K Groves http://orcid.org/0000-0002-0784-7998

### Ethics

This study was performed in strict accordance with the recommendations in the Guide for the Care and Use of Laboratory Animals of the National Institutes of Health. All of the animals were handled according to approved institutional animal care and use committee (IACUC) protocol (AN4956) of Baylor College of Medicine.

Decision letter and Author response
Decision letter https://doi.org/10.7554/eLife.79712.sa1
Author response https://doi.org/10.7554/eLife.79712.sa2

## Additional files

### Supplementary files
• MDAR checklist

### Data availability
Sequencing data have been deposited in GEO under accession codes GSE182202.

The following dataset was generated:

| Author(s) | Year | Dataset title | Dataset URL | Database and Identifier |
| --- | --- | --- | --- | --- |
| Groves AK, Iyer AA, Hosamani I | 2022 | Single cell transcriptomic analysis of P8 and P15 mouse cochlea (control and three overexpression conditions) | https://www.ncbi.nlm.nih.gov/geo/query/acc.cgi?acc=GSE182202 | NCBI Gene Expression Omnibus, GSE182202 |

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
