## [Editor Report]

This study uses well-designed genetic approaches to examine how specific transcription factors regulate hair cell regeneration that could have implications for hearing loss. Although it was felt there could be more functional electrophysiology assessments we appreciate that this is beyond the current capability of the lab.

---

## [Decision Letter]

**Decision letter after peer review:**

Thank you for submitting your article "Cellular reprogramming with ATOH1, GFI1, and POU4F3 implicate epigenetic changes and cell-cell signaling as obstacles to hair cell regeneration in mature mammals" for consideration by *eLife*. Your article has been reviewed by 2 peer reviewers, and the evaluation has been overseen by a Reviewing Editor and Mone Zaidi as the Senior Editor. The reviewers have opted to remain anonymous.

Essential revisions:

1) Please show functional characterization of reprogrammed hair cells in P8-P15 mice.

2) Please address why the authors chose not to explore ATOH-1 and Pou4F3 without GFl-1.

*Reviewer #1 (Recommendations for the authors):*

In this manuscript, Iyer et al., investigated the effect of combining two transcription factors of promoting the reprogramming potential of Atoh1 in newborn mice and older mice. In neonatal mice, the authors reported that Atoh1 overexpression is sufficient to drive highly efficient reprogramming of non-sensory cochlear tissue into hair cell-like cells which showed FM1-43 intake. However, in older mice (>1 week), overexpression of additional transcription factors GFI1 and POU4F3 is required to enhance Atoh1-induced reprogramming of supporting cells into hair cells, which was supported by RNAseq analysis showing reprogramed cells under these conditions are transcriptionally similar and possess immature hair cell-like features. Furthermore, they showed that Notch signaling likely counteracts the reprogramming effect of these transcription factors and further demonstrated that hair cell loci become less epigenetically accessible in supporting cells and GER cells between postnatal days 1 and 8. These findings provide mechanistic insights into why Atoh1 and associated transcription factors have less reprogramming potential in older mice. Overall, the experiments were well justified and designed. However, besides showing uptake of FM1-43 in reprogrammed hair cells in neonatal mice, no functional characterization of reprogrammed hair cells in P8-P15 mice. It is possible that not only do older mice have less epigenetic accessibility in supporting cells and GER cells, but the reprogrammed hair cells in older mice might also have limited functionality. This needs to be tested. Along the line, while P8 reprogrammed hair cells show robust enrichment in genes related to sensory perception of sound (Figure 3E), it is the cell projection in P15 reprogrammed hair cells (Figure 6E). Moreover, the changes in transduction channel protein expression in neonatal, P8, and P15 reprogrammed hair cell gene sets should be analyzed. In addition, it will be a plus to characterize the electrophysiological properties of these reprogrammed hair cells at different stages.

*Reviewer #2 (Recommendations for the authors):*

1. The authors' comment that using Rosa26 as a promoter helps the genetic design to circumvent the issue of multiple copy numbers of the transgene is factually incorrect. However, the CRISPR-Cas9 system has a caveat of having a high chance of random insertion(s) and/or deletion(s) of the transgene and the native genome. Since the authors did not directly assess the copy number of the transgene, they should avoid this claim.

---

## [Author Response]

Essential Revisions (for the authors):1) Please show functional characterization of reprogrammed hair cells in P8-P15 mice.

We have now performed FM1-43 labeling of the reprogrammed cells in our mice that were reprogrammed at P8 and analyzed at P15. Because the temporal bone is calcified at P15, it is hard to expose the cochlear epithelium for rapid live imaging. We therefore isolated the cochlea, perfused a fixable form of FM1-43 into the cochlea and immediately washed it out with buffer followed by fixative. Our data show that this method labels the endogenous organ of Corti hair cells very well, and we also see a small amount of FM1-43 labeling in the ectopic reprogrammed hair cells in our mice. These data are now included as Figure 5 – Supplementary Figure 1. This suggests that the reprogrammed hair cells retain at least rudimentary mechanotransduction properties at this age, but confirms our electron microscopy data, antibody marker data and scRNA-seq data that reprogramming at this later age produces cells that are less mature than the reprogrammed cells generated at earlier ages.

2) Please address why the authors chose not to explore ATOH-1 and Pou4F3 without GFl-1.

As the reviewer correctly states, previous work from both vertebrates and *Drosophila* suggest that Atoh1 and Gfi1 can synergize to enhance the transcriptional activity of Atoh1. We recently showed that Gfi1 does this by forming a complex with Atoh1 where Gfi1 does not itself bind DNA (Jen et al., 2022). A study from the Jarman lab (unpublished but on BioRxiv) also shows that Gfi1 is critical for the expression of hair cell genes when combined with Atoh1 and Pou4f3. Finally, we and our collaborators have also shown that viral over-expression of Gfi1 enhances the hair cell reprogramming ability of Atoh1 in vivo (Lee et al., 2020). This is why we chose to include Gfi1 in our series of reprogramming mice rather than simply test Atoh1 and Pou4f3 together.

Although we expected Gfi1 to enhance the activity of Atoh1 alone, we point out that our reprogramming experiments were only analyzed after one week. It is possible that at longer times we would see additional hair cell genes up-regulated in the Gfi1+Atoh1 condition compared to Atoh1 alone. Indeed our previous study (Lee et al., 2020) demonstrated Gfi1 enhances Atoh1 activity when both factors are over-expressed using adenovirus in deafened adult mice after 4 or 8 weeks.

Reviewer #1 (Recommendations for the authors):In this manuscript, Iyer et al., investigated the effect of combining two transcription factors of promoting the reprogramming potential of Atoh1 in newborn mice and older mice. In neonatal mice, the authors reported that Atoh1 overexpression is sufficient to drive highly efficient reprogramming of non-sensory cochlear tissue into hair cell-like cells which showed FM1-43 intake. However, in older mice (>1 week), overexpression of additional transcription factors GFI1 and POU4F3 is required to enhance Atoh1-induced reprogramming of supporting cells into hair cells, which was supported by RNAseq analysis showing reprogramed cells under these conditions are transcriptionally similar and possess immature hair cell-like features. Furthermore, they showed that Notch signaling likely counteracts the reprogramming effect of these transcription factors and further demonstrated that hair cell loci become less epigenetically accessible in supporting cells and GER cells between postnatal days 1 and 8. These findings provide mechanistic insights into why Atoh1 and associated transcription factors have less reprogramming potential in older mice. Overall, the experiments were well justified and designed. However, besides showing uptake of FM1-43 in reprogrammed hair cells in neonatal mice, no functional characterization of reprogrammed hair cells in P8-P15 mice. It is possible that not only do older mice have less epigenetic accessibility in supporting cells and GER cells, but the reprogrammed hair cells in older mice might also have limited functionality. This needs to be tested.

As noted above, we have now repeated these experiments in P8-P15 mice and have shown that the reprogrammed hair cells can indeed take up FM1-43. These data are now included as Figure 5 – Supplementary Figure 1.

Along the line, while P8 reprogrammed hair cells show robust enrichment in genes related to sensory perception of sound (Figure 3E), it is the cell projection in P15 reprogrammed hair cells (Figure 6E). Moreover, the changes in transduction channel protein expression in neonatal, P8, and P15 reprogrammed hair cell gene sets should be analyzed.

We have now analyzed our scRNA-seq data at both P8 and P15 time points and have analyzed expression of components of the mechanotransduction apparatus in the reprogrammed cells: the two putative channels TMC1 and 2, their accessory transmembrane proteins TMIE and LHFPL5/TMHS and the two components of the tip links, CDH23 and PCDH15. We find that TMC1, TMIE, LHFPL5, CDH23 and PCDH15 can all be detected in P8 reprogrammed cells compared to their unreprogrammed counterparts. At P15, we see less expression of the channel components, consistent with the more immature morphology of the reprogrammed hair cells at this age and the lower amount of FM1-43 labeling. These violin plots are now included as Figure 6 – Supplementary Figure 2.

In addition, it will be a plus to characterize the electrophysiological properties of these reprogrammed hair cells at different stages.

Our lab does not have electrophysiology expertise and so we have not been able to perform these experiments. We hope, however, that our FM1-43 labeling and scRNA-seq data go some way to demonstrating rudimentary mechanotransduction properties of the new hair cells.

Reviewer #2 (Recommendations for the authors):1. The authors' comment that using Rosa26 as a promoter helps the genetic design to circumvent the issue of multiple copy numbers of the transgene is factually incorrect. However, the CRISPR-Cas9 system has a caveat of having a high chance of random insertion(s) and/or deletion(s) of the transgene and the native genome. Since the authors did not directly assess the copy number of the transgene, they should avoid this claim.

It is well-established that knock-in mice created by homologous recombination to a defined genomic location have fewer copy numbers (i.e. one copy) than mice generated by transgenesis. Nevertheless, we should have made more clear that we were not generating transgenic mice using the ROSA26 promoter in this study – we targeted the endogenous ROSA26 locus by homologous recombination. Moreover, we maintained our targeted mouse lines in the homozygous state by genotyping for the presence of the modified ROSA26 allele and the absence of the wild type ROSA26 allele. This suggests our targeting is specific to the ROSA26 locus. While it is formally possible that an extra intact copy of the targeting vector integrated elsewhere in the mouse genome, the out-crossing required to expand each founder line would tend to eliminate such modified alleles when selecting for a targeted ROSA modification at the endogenous ROSA locus on chromosome 6.